# PERSONALIZATION MITIGATES THE PERILS OF LOCAL SGD FOR HETEROGENEOUS DISTRIBUTED LEARNING

## ABSTRACT

This paper investigates a personalized version of Local Stochastic Gradient Descent (Local SGD). We establish improved convergence guarantees for this personalized approach, eliminating the need for extra assumptions about data or gradient heterogeneity. Our theoretical analysis reveals that personalized Local SGD outperforms both pure local training and federated learning algorithms that produce a consensus model for all devices. This performance gain is primarily due to over-parameterization, which allows for reducing the consensus error between clients with more communication—something that is not observed in non-personalized approaches. We illustrate our observations using experiments on synthetic convex and smooth objectives.

## 1 INTRODUCTION

Collaborative machine learning protocols have been instrumental in driving recent scientific breakthroughs (Bergen & Petryshen, 2012). These protocols are increasingly being adopted across a variety of sectors: from networks of hospitals (Li et al., 2019; Powell, 2019; Roth et al., 2020) and mobile devices (McMahan & Ramage, 2017; Apple; Paulik et al., 2021), to the banking industry (Shiffman et al., 2021), and have even proven valuable in studying COVID-19 (Dayan et al., 2021). Most collaborative learning problems with $M$ machines/participants can be stated as multi-criterion optimization problems of the following form:

$$\min_{v_1,\ldots,v_M \in \mathcal{W}} (F_1(v_1),\ldots,F_M(v_M)) \quad, \tag{1}$$

where $F_m(v) = \mathbb{E}_{z \sim \mathcal{D}_m}[f(v;z)]$ is the objective of machine $m$ defined using a data distribution $\mathcal{D}_m$ and a loss function $f : \mathcal{W} \times \mathcal{Z} \to \mathbb{R}$. Machine $m$ can solve its problem locally, i.e., without collaboration, if (i) it can fully access its objective $F_m$ through unrestricted access to $\mathcal{D}_m$ and (ii) it does not have computational/time constraints. If these two assumptions hold for all machines, Problem (1) degenerates into $M$ different optimization problems. However, at least one, and in most cases, both of these assumptions fail in practice. For instance, assume each machine can only access a data set, $S_m \sim \mathcal{D}_m^{\otimes T}$, with $|S_m| = T$ where $T$ is much smaller than the sample complexity to optimize $F_m$ to some target sub-optimality $\epsilon$. Or even in the online setting, where at each time step, the machine $m$ gets a sample $z_t^m \sim \mathcal{D}_m$, the time complexity to reach a good solution might be too high. As a result, using pure local training to obtain a good model can be prohibitive in the worst case and very expensive in the best case.

Fortunately, in many real applications such as next-word prediction on a mobile keyboard (Hard et al., 2018), these $M$ objectives/distributions share several similarities, and sharing information between the machines can drastically cut the total training time and sample complexity (Blum et al., 2017; Haghtalab et al., 2022). This is the motivation behind the rapidly growing field of federated learning (FL) (McMahan et al., 2016b;a; Kairouz et al., 2019), that usually simplifies Problem (1) in two steps: first, by using a **consensus model** for all the participants,

$$\min_{v \in \mathcal{W}} (F_1(v),\ldots,F_M(v)) \quad, \tag{2}$$

and then by **linearly scalarizing** the objective to get a simple optimization problem,

$$\min_{v \in \mathcal{W}} \left( F(v) := \frac{1}{M} \sum_{m \in [M]} F_m(v) \right) \quad. \tag{3}$$

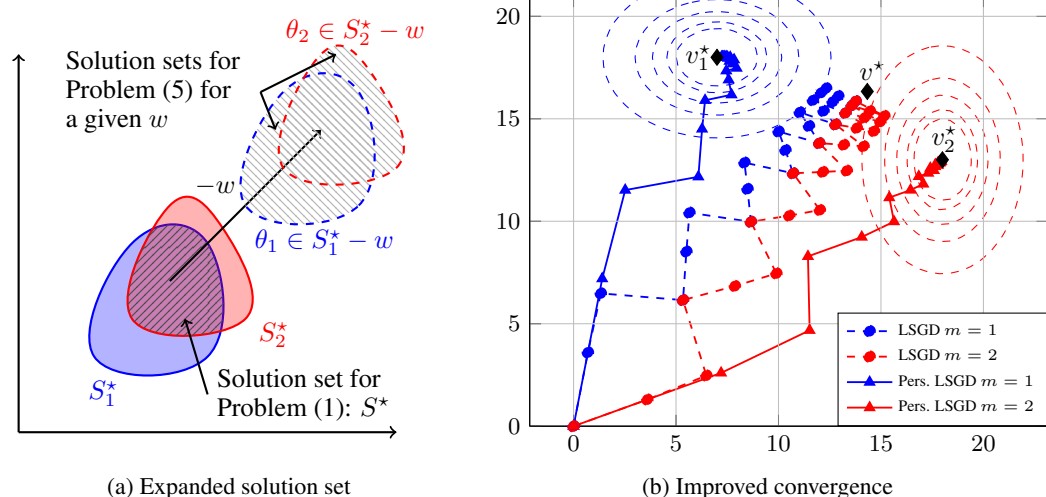

(a) Expanded solution set          (b) Improved convergence

Figure 1: **The benefit of over-parameterization offered by personalization.** (a) An illustration of how the solution concept changes by incorporating additive personalization as in Problem (5). Without personalization, the solution of Problem (3) must be in the set $\cap_{m\in[M]} S_m^\star$, which can be a very small set, when the number of machines is large. On the other hand, with additive personalization, for every shared model $w$, $\theta_m$ can be in the translated set $S_m^\star - w$. This significantly expands the solution set for Problem (5). See Section 3.3 for more discussions. (b) Optimization trajectories of local GD (w/ $\eta = 0.05$) v/s personalized local GD (w/ $\eta = 0.05$, $\alpha = 1$) with $\mathcal{W} = \mathbb{R}^2$, $M = 2$, $K = 3$ and $R = 8$. The blue machine has objective $F_1(v) = (v[1] - 7)^2 + 2(v[2] - 18)^2 - 1$, while the red machine has objective $F_2(v) = 2(v[1] - 18)^2 + (v[2] - 13)^2 - 1$. We denote contour lines for both functions in their respective colors. $v_1^\star$ and $v_2^\star$ are the optima of the machines respectively, while $v^\star$ is the optimum of the average function $F(v) = (F_1(v) + F_2(v))/2$.

In low heterogeneity settings, where the data distributions are similar, Problem (3) is a good proxy for Problem (1). This is why much effort has been put into studying optimization algorithms for Problem (3) [1]. In the extreme case when $\mathcal{D}_1 = \cdots = \mathcal{D}_M$, i.e., the homogeneous setting, we know the min-max complexity of smooth first-order optimization (Woodworth et al., 2021; Patel et al., 2022) as well as tight convergence guarantees for the most popular federated optimization algorithm, i.e., *local SGD/FedAvg* (Woodworth et al., 2020a; Glasgow et al., 2022).

However, in the heterogeneous setting, **(I)** it is unclear if Problem (3) remains a reasonable proxy to Problem (1) as the averaged objective function might be far away from local ones. And even if Problem (3) is a good proxy, **(II)** strong heterogeneity assumptions (Haddadpour & Mahdavi, 2019; Khaled et al., 2020; Woodworth et al., 2020b) are needed to analyze local SGD, or we cannot theoretically demonstrate its practical effectiveness (Koloskova et al., 2020; Wang et al., 2022; Patel et al., 2023) in optimizing Problem (3). This is because local update methods suffer from *"client drift"* between communication rounds (Zhao et al., 2018; Charles & Konecny, 2020; Karimireddy et al., 2020) as the client devices optimize their own objective $F_m$ instead of the averaged objective $F$ (c.f., Figure 1b). This client drift is known to slow down the convergence of the local update algorithms for Problem (3) (Glasgow et al., 2022; Patel et al., 2023).

To alleviate issues **(I)** and **(II)**, we take forward (Arivazhagan et al., 2019; Liang et al., 2020; Hanzely et al., 2021; Bietti et al., 2022; Pillutla et al., 2022; Mishchenko et al., 2023) the study of the following *personalization-aware scalarization* of Problem (1):

$$\min_{\psi := (w, \theta_1, \ldots, \theta_M) \in \mathcal{W}^{M+1}} \left( \hat{F}(\psi) := \frac{1}{M} \sum_{m \in [M]} F_m(g(w, \theta_m)) \right) \ , \tag{4}$$

---

[1]Besides providing an approximate solution for Problem (1), there are also other motivations to output a single consensus model, such as learning a foundation model for other downstream tasks (Zhuang et al., 2023).

where $g : \mathcal{W}^2 \to \mathcal{W}$ is an *"aggregation rule"* that specifies how to combine the shared *"global model"* $w \in \mathcal{W}$ and a *"client-specific model"* $\theta_m \in \mathcal{W}$. Hanzely et al. (2021) have shown that Problem (4) recovers several personalized federated learning (PFL), multi-task learning, and meta-learning formulations. Furthermore, Mishchenko et al. (2023) have empirically shown that this *"partial personalization"* effectively leverages the shared structure across machines for several deep learning tasks. In this paper, we study the simplest and yet, as we demonstrate, powerful, additive aggregation rule for Problem (4), with $g(w, \theta) \coloneqq w + \theta$. This results in the following problem:

$$\min_{\psi \coloneqq (w, \theta_1, \ldots, \theta_M) \in \mathcal{W}^{M+1}} \left( \hat{F}(\psi) \coloneqq \frac{1}{M} \sum_{m \in [M]} F_m(w + \theta_m) \right) \ . \tag{5}$$

The additive aggregation makes the problem under-determined for each machine and is tantamount to directly scalarizing problem 1. For example, consider a supervised learning task like product recommendation, where $F_m(w + \theta_m) = \mathbb{E}_{(x,y) \sim \mathcal{D}_m} [l(y, \langle w + \theta, x \rangle)]$. Here, $l$ represents a loss function such as logistic loss, $x$ is the feature vector for a product, and $y$ is a binary label indicating a customer's purchase. In such cases, there exist common population-level features like average product ratings and technical specifications, as well as machine-specific features such as budget and color preference. The additive model enables collaboration to enhance the model for shared features while maintaining the capability for user-specific customization. Due to its simplicity and this modularity, the additive model has recently been studied, including by Bietti et al. (2022) in convex settings and as a specialized case of robust aggregation by Pillutla et al. (2022). We propose Algorithm 1, a variant of personalized local SGD generalizing both (i) pure local training and (ii) vanilla local SGD to solve Problem (5). Our contributions are as follows:

- In the strongly convex setting (see Section 3), we identify the regime where Algorithm 1 strictly improves over both algorithmic extremes (i) and (ii), attaining the best possible rate for any non-accelerated collaborative first-order algorithm. We make no data heterogeneity assumptions in our analysis, which is a significant improvement over every known vanilla local SGD analysis.

- In the general convex setting (see Section 3.2), we provide a convergence analysis in terms of consensus error: a key quantity that appears in all local SGD analyses (Koloskova et al., 2020; Khaled et al., 2020; Woodworth et al., 2020b). Using a theoretical hard instance, we show that while the consensus error of vanilla local SGD between communication rounds stays constant, it is driven to zero with personalization (c.f., Figure 2).

- We highlight that the benefit of personalization in the additive model comes from over-parameterization (see Section 3.3). Over-parameterization expands the solution set of the underlying optimization problem and, thus, makes it easier for the local-update method to converge to the correct optima–a benefit that is crucial with higher data heterogeneity (c.f., Figure 1).

**Notation.** We use $\preceq, \equiv, \succeq$ to denote scalar equalities and inequalities up to numerical constants. When comparing positive semi-definite matrices, we use $\preceq, \succeq$ to denote the Loewner order. We will consider unconstrained optimization over the $d$-dimensional Euclidean space, i.e., $\mathcal{W} = \mathbb{R}^d$. For $v \in \mathbb{R}^d$, $\|v\|$ denotes the associated Euclidean norm and $v[i]$ denotes the $i^{th}$ co-ordinate of $v$. For a matrix $A \in \mathbb{R}^{d \times d}$, $\lambda_{max}(A), \lambda_{min}(A)$ denote its largest and smallest eigen-values, and $\|A\|_2 = \lambda_{max}(A)$ denotes the spectral norm of the matrix.

## 2 SETTING AND PRELIMINARIES

**Function class.** We assume each machine's objective is (strongly) convex and smooth.

**Assumption 1** (Regularity assumptions). *All objective functions $F_m$ are twice-differentiable, $L$-smooth, and $\mu$-strongly convex, where $0 \leq \mu \leq L$. We recall the following equivalent properties on the Hessian*

$$\mu \cdot I_d \preceq \nabla^2 F_m(v) \preceq L \cdot I_d, \forall \, v \in \mathcal{W} \ , \tag{6}$$

*on the gradient bounds*

$$\mu \|v_1 - v_2\| \leq \|\nabla F_m(v_1) - \nabla F_m(v_2)\| \leq L \|v_1 - v_2\| , \ \forall v_1, v_2 \in \mathcal{W} \ , \tag{7}$$

*and on the function bounds*

$$\frac{\mu}{2} \|v_1 - v_2\|^2 \leq F_m(v_1) - F_m(v_2) - \langle \nabla F_m(v_2), v_1 - v_2 \rangle \leq \frac{L}{2} \|v_1 - v_2\|^2 , \ \forall v_1, v_2 \in \mathcal{W} \ . \tag{8}$$

| Algorithm | Optimization Terms | Noise Terms | Heterogeneity Terms |
|---|---|---|---|
| **Strongly Convex Setting:** $\frac{1}{M}\sum_{m\in[M]}\mathbb{E}\left[\|\hat{v}_m - v_m^\star\|^2\right] \preceq$ | | | |
| MB-SGD (Dekel et al., 2012) | $B^2 e^{-\frac{R}{\kappa}}$ | $\frac{\sigma^2}{\mu^2 MKR}$ | $\frac{\zeta_\star^2}{L^2}$ |
| Local SGD (Koloskova et al., 2020) | $\kappa B^2 e^{-\frac{R}{\kappa}}$ | $\frac{\sigma^2}{\mu^2 MKR} + \frac{\kappa\sigma^2}{\mu^2 KR^2}$ | $\zeta_\star^2 \cdot \left(\frac{1}{L^2} + \frac{\kappa}{\mu^2 R^2}\right)$ |
| Pure Local Training | $B^2 e^{-\frac{KR}{2\kappa}}$ | $\frac{\sigma^2}{\mu^2 KR}$ | - |
| Algorithm 1 (**Theorem** 1, (16)) | $B^2 e^{-\frac{KR}{2\kappa}}$ | $\frac{\sigma^2}{\mu^2 MKR}$ | - |
| **Convex Setting:** $\frac{1}{M}\sum_{m\in[M]} F_m(\hat{v}_m) - F_m^\star \preceq$ | | | |
| MB-SGD (Dekel et al., 2012) | $\frac{LB^2}{R}$ | $\frac{\sigma B}{\sqrt{MKR}}$ | $\frac{\zeta_\star^2}{L}$ |
| Local SGD (Koloskova et al., 2020) | $\frac{LB^2}{R}$ | $\frac{\sigma B}{\sqrt{MKR}} + \frac{(L\sigma^2 B^4)^{1/3}}{K^{1/3}R^{2/3}}$ | $\frac{\zeta_\star^2}{L} + \frac{(L\zeta_\star^2 B^4)^{1/3}}{R^{2/3}}$ |
| Pure Local Training | $\frac{LB^2}{KR}$ | $\frac{\sigma B}{\sqrt{KR}}$ | - |
| Algorithm 1 (Hanzely et al., 2021) | $\frac{MLB^2}{R}$ | $\frac{\sigma B}{\sqrt{KR}} + \frac{(L\sigma^2 M^2 B^4)^{1/3}}{R^{2/3}}$ | $\frac{(L\zeta_\star^2 M^2 B^4)^{1/3}}{R^{2/3}}$ |
| Algorithm 1 (w/ $K=1$) (Bietti et al., 2022) | $\frac{LB^2}{R}$ | $\frac{\sigma B}{\sqrt{MKR}}$ | - |
| Algorithm 1 (**Conjecture** 1) | $\frac{MLB^2}{KR}$ | $\frac{\sigma B}{\sqrt{MKR}} + \chi_1(\sigma, K, R)$ | $\chi_2(\zeta_\star, R)$ |

Table 1: Summary of exiting convergence rates in the (strongly) convex setting. The guarantees for all non-personalized algorithms are translated to personalized guarantees using Assumption 3. Pure local training refers to running SGD separately on each machine. Thus, its guarantee is standard and can be found in Stich (2019). Bietti et al. (2022)'s and our guarantees are for additive personalization. Furthermore, our guarantee is stated in the regime where $K < \frac{K}{2(1-e^{-1/4MR})} \leq \kappa < 4MKR$, as highlighted in Theorem 1. In the convex setting, due to Patel et al. (2023), local SGD is dominated by mini-batch SGD, but we include it for completeness. For the conjectured rate for Algorithm 1 in the convex setting, see Section 3.2.

We recall that there is a unique optimizer for strictly convex functions, i.e., when $\mu > 0$. In the strictly convex setting, we denote the condition number of the function by $\kappa = \frac{L}{\mu}$. When $\mu = 0$, we call our functions just convex. Next, we assume the machines' optima are bounded. In particular, we denote the sets of optima of the average function and each machine as follows:

$$S^\star := \arg\min_{v\in\mathcal{W}} F(v) \text{ and } S_m^\star := \arg\min_{v\in\mathcal{W}} F_m(v), \forall m \in [M] \ . \tag{9}$$

**Assumption 2** (Bounded local solutions). *There exists $B > 0$, such that for every solution $v_m^\star \in S_m^\star$, we have $\|v_m^\star\| \leq B$.*

We will also use the following heterogeneity assumption in discussing guarantees of related algorithms. Note that we **do not** require the assumption to analyze Algorithm 1 (c.f., Theorem 1).

**Assumption 3** (Bounded heterogeneity at optima). *Let $F_m$'s satisfy Assumption 1. For a given machine $m$, we define $d(S_m^\star, S^\star) := \arg\max_{v_m^\star \in S_m^\star, v^\star \in S^\star} \|v_m^\star - v^\star\|_2^2$. Then, for given $\zeta_\star \geq 0$,*

$$\frac{1}{M} \sum_{m\in[M]} d(S_m^\star, S^\star) \leq \frac{\zeta_\star^2}{L^2} \ . \tag{10}$$

**Remark.** *For L-smooth functions, this assumption implies another less restrictive assumption which has been used in several works both without (Koloskova et al., 2020; Woodworth et al., 2020b; Patel*

*et al., 2022)* and with personalization *(Hanzely et al., 2021)* for analyzing local update methods in the (strongly) convex setting. In particular, for all $v^\star \in S^\star$ and $v_m^\star \in S_m^\star$,

$$\frac{1}{M} \sum_{m \in [M]} \|\nabla F_m(v^\star)\|^2 = \frac{1}{M} \sum_{m \in [M]} \|\nabla F_m(v^\star) - \nabla F_m(v_m^\star)\|^2 \overset{(7)}{\leq} \frac{L^2}{M} \sum_{m \in [M]} \|v_m^\star - v^\star\|^2 \overset{(10)}{\leq} \zeta_\star^2 \ .$$

**Oracle and communication model.** One key benefit of using the additive model for personalization as introduced in Problem (5) is that,

$$\nabla_w F_m(w + \theta) = \nabla_\theta F_m(w + \theta) = \nabla_{w+\theta} F_m(w + \theta) \ .$$

In particular, the gradients on machine $m$ w.r.t. its copies of the global and personal models $w_t^m, \theta_t^m$ at time $t$ are only a function of their sum, $v_t^m := w_t^m + \theta_t^m$. This means we do not need to compute multiple gradients at any time step despite using a stronger personalized model (c.f., line 6 in Algorithm 1) and allows us to consider the following standard stochastic oracle on each machine.

**Definition 1.** *Each machine has access to a stochastic first-order oracle $\mathcal{O}_m : \mathcal{W} \rightarrow \mathcal{W}$ which outputs $\mathcal{O}_m(v_t^m) = g_t^m$ at time $t \in [T]$ such that, $\mathbb{E}[g_t^m | v_t^m] = \nabla F_m(v_t^m)$ and, $Var(g_t^m | v_t^m) \leq \sigma^2$, where we define $Var(u|v) := \mathbb{E}\left[ \|u - \mathbb{E}[u|v]\|^2 | v\right].$*

In our setting, the above oracle can be implemented simply by first sampling a new data point $z_t^m \sim \mathcal{D}_m$ on machine $m$ at time $t$ and then returning $\nabla f(v_t^m; z_t^m)$ for the query $v_t^m$. Finally, to model the expensive nature of communication in collaborative learning *(Wang et al., 2021)*, we will consider the *"intermittent communication model"* *(Woodworth et al., 2018; Woodworth, 2021)*, illustrated in Figure 3 in Appendix A. In particular, there are $T = KR$ time steps, and the devices communicate $R$ times with $K$ time steps in between. Thus, communication happens before generating models with index in $\{K, 2K, \ldots, K(R-1), KR\}$.

**Non-personalized to personalized guarantees.** As discussed in Section 1, most optimization algorithms in federated learning solve Problem (3) while personalized FL algorithms aim to minimize the following objective, which is a scalarization of problem (1),

$$\min_{v_1,\ldots,v_M \in \mathcal{W}} \frac{1}{M} \sum_{m \in [M]} F_m(v_m) \ , \tag{11}$$

which is implicit in Problem (4). But we can still facilitate a comparison between personalized and non-personalized guarantees. In particular, in the convex setting we bound the following *"cost of not personalizing"*:

$$\min_{v^\star \in \mathcal{W}} \frac{1}{M} \sum_{m \in [M]} F_m(v^\star) - \min_{v_1^\star,\ldots,v_M^\star \in \mathcal{W}} \frac{1}{M} \sum_{m \in [M]} F_m(v_m^\star) \ . \tag{12}$$

This is precisely the excess loss incurred by any algorithm that insists on using a consensus model for all the devices. First, using $L$-smoothness of $F_m$ followed by Assumption 3, we can show that this cost is upper bounded by $\zeta_\star^2/L$. Furthermore, there is a convex problem instance (see proof of Proposition 1) satisfying Assumptions 1, 2 and with $\frac{1}{M} \sum_{m \in [M]} \|v^\star - v_m^\star\|^2 = \frac{\zeta_\star^\star}{L^2}$ (i.e., it also satisfies Assumption 3) such that Equation (11) is lower bounded by $\frac{\zeta_\star^2}{L}$. Thus, using an algorithm for solving Problem (3), we can not hope to do better than $\frac{\zeta_\star^2}{L}$ in terms of function sub-optimality. Similarly, in the strongly convex setting, since we are interested in bounding the average of the distance to the optimum $v_m^\star$ on each machine (as opposed to function sub-optimality), the cost of not personalizing is given by

$$\frac{1}{M} \sum_{m \in [M]} \|v^\star - v_m^\star\|^2 , \tag{13}$$

which is directly bounded by $\frac{\zeta_\star^2}{L^2}$ using Assumption 3. This upper bound is also tight (assuming $\zeta_\star^2 \leq 2L^2B^2$) for problems satisfying Assumptions 1, 2 and 3. We use these reductions in Table 1.

## 3 PERSONALIZED LOCAL SGD ALGORITHM AND ITS ANALYSIS

---

**Algorithm 1** Local SGD with personalization

---
1: **Input:** inner step-size sequence $\eta_t$, outer step-size $\beta$, personalization parameter $\alpha$, initializations $w_0, \theta_0$
2: Initialize $w_0^m = w_0$, $\theta_0^m = \theta_0$ on all machines $m \in [M]$
3: **for** $t \in \{0, \ldots, KR - 1\}$ **do**
4:     **for** $m \in [M]$ **in parallel do**
5:         Sample $z_t^m \sim \mathcal{D}_m$
6:         Compute the stochastic gradient $\nabla f(w_t^m + \theta_t^m; z_t^m)$
7:         $\theta_{t+1}^m \leftarrow \theta_t^m - \alpha\eta_t\nabla f(w_t^m + \theta_t^m; z_t^m)$
8:         $w_{t+1}^m \leftarrow w_t^m - \eta_t\nabla f(w_t^m + \theta_t^m; z_t^m)$
9:         **if** $(t + 1)\bmod K = 0$ **then**
10:             **Communicate** $w_{t+1}^m$ to server
11:             **Server makes update** $w_{t+1} \leftarrow w_{t+1-K} + \beta\frac{1}{M}\sum_{m'\in[M]}\left(w_{t+1}^{m'} - w_{t+1-K}\right)$
12:             **Receive** $w_{t+1}$ from server
13:             $w_{t+1}^m \leftarrow w_{t+1}$
14:         **end if**
15:     **end for**
16: **end for**
17: **Output:** $\hat{\phi} := \left(w_T + \theta_T^1, \ldots, w_T + \theta_T^M\right)$          ▷ Option I

        $\hat{\phi} := \frac{1}{TM}\sum_{t\in[T],m\in[M]}\left(w_t^m + \theta_t^1, \ldots, w_t^m + \theta_t^M\right)$      ▷ Option II

        $\hat{\phi} := \frac{1}{T}\sum_{t\in[T]}\left(w_t^1 + \theta_t^1, \ldots, w_t^M + \theta_t^M\right)$          ▷ Option III

---

In Algorithm 1, we showcase a personalized local SGD algorithm that differs from vanilla local SGD (c.f., Algorithm 2) because of the introduction of the machine-specific model $\theta_t^m$, and the personalization parameter $\alpha$. Varying $\alpha$ w.r.t. the outer step-size $\beta$ controls the personalization of the final models. As long as $\alpha > 0$, our algorithm will have some personalization. Specifically, we progressively reduce the personalization rate by increasing the ratio $\beta/\alpha$ and appropriately scaling $\eta$ to ensure that the algorithm converges. We can note the following two extreme cases:

- When $\alpha = 0$, Algorithm 1 recovers the familiar local SGD algorithm with an inner-outer step-size which has been discussed in several works such as Karimireddy et al. (2020); Charles & Konecny (2020); Wang et al. (2022). Further setting $\beta = 1$ recovers vanilla local SGD as analyzed in Woodworth et al. (2020b); Yuan & Ma (2020). And instead, setting $\eta_t = \eta = 0$ recovers large mini-batch SGD with batch-size $MK$ and step-size $\beta$.

- When $\beta = 0$, $\eta_t = \eta$ tends to zero, and $\alpha\eta$ is a constant, Algorithm 1 recovers pure local training, i.e., SGD on each machine with step-size $\alpha\eta$. We show in Theorems 1 and 2 that Algorithm 1 can even recover the rate of pure local training without being in this limiting regime.

An initial version of Algorithm 1 was presented by Arivazhagan et al. (2019); Liang et al. (2020), emphasizing the empirical advantages of using global and local models. However, their work lacks optimization results. A theoretical analysis was later provided by Hanzely et al. (2021), who explored the convergence aspects of Algorithm 1 for a general aggregation function with $\alpha = 1$. Unfortunately, their approach included unconventional assumptions, like a unique optimal point for general convex functions. Even with these assumptions, their results are dominated by pure-local training, as indicated in Table 1. Their strongly convex analysis is also not fully worked out, and they rely on Assumption 3, while our analysis does not.

Building upon the insights from Bietti et al. (2022)—whose study on Algorithm 1 in the general convex scenario (without local updates) is notable and offers better rates than Hanzely et al. (2021) (c.f., Table 1)—our analyses go further. We provide a faster rate in the strongly convex setting using local update steps in Theorem 1. Our exploration, as detailed in Section 3.2, reveals that personalization mitigates the persistent consensus error due to data heterogeneity. This positive

impact of personalization can be bestowed to over-parameterization, as elaborated in Section 3.3 and supported by recent empirical findings from Mishchenko et al. (2023).

## 3.1 THE STRONGLY CONVEX SETTING

The discussion above highlights that with specific settings of hyper-parameters, Algorithm 1 can recover the guarantees of both the extremes: (i) pure local training with SGD and (ii) the most popular federated learning algorithms, local and MB-SGD. **But is it possible to show that in some regimes, Algorithm 1 is also provably strictly better than both these extremes?** We show that this is indeed possible in the following convergence result.

**Theorem 1** (Strongly Convex Functions). *Assume the functions on each machine satisfy Assumptions 1 (with $\mu > 0$) and 2. Define the condition number of the problems as $\kappa = \frac{L}{\mu}$. For each machine $m$, we output $\hat{v}_m := w_{KR} + \theta_{KR}^m$, i.e., the final model on each machine $m \in [M]$ after the $R^{th}$ communication round ( Option I in Algorithm 1). Then we can get the following guarantees for Algorithm 1 when $\eta_t = \eta < \frac{1}{(1+\alpha)L}$ and $w_0 = \theta_0 = 0$,*

$$\frac{1}{M} \sum_{m \in [M]} \mathbb{E} \|\hat{v}_m - v_m^\star\|_2^2 \le e^{-\frac{2\alpha R}{1+\alpha}\left(1 - e^{-\eta(1+\alpha)\mu K}\right)} B^2 + \frac{4\eta\sigma^2\alpha}{\mu} \cdot \left(1 + \frac{\beta^2}{M\alpha^2}\right) \ . \quad (14)$$

*Furthermore, setting $\beta \le \sqrt{M}$ and $\alpha = 1$, we get that for $\eta < \frac{1}{2L}$,*

$$\frac{1}{M} \sum_{m \in [M]} \mathbb{E} \|\hat{v}_m - v_m^\star\|_2^2 \le e^{-R\left(1 - e^{-2\eta\mu K}\right)} B^2 + \frac{8\eta\sigma^2}{\mu} \ . \quad (15)$$

*Finally, assuming $K < \frac{K}{2(1 - e^{-1/4MR})} \le \kappa < 4MKR^2$ and setting $\eta = \frac{1}{8\mu MKR}$, we obtain that,*

$$\frac{1}{M} \sum_{m \in [M]} \mathbb{E} \|\hat{v}_m - v_m^\star\|_2^2 \le e^{-\frac{KR}{2\kappa}} B^2 + \frac{\sigma^2}{\mu^2 MKR} \ . \quad (16)$$

We can make several observations about the result presented in Theorem 1. Firstly, from bound (14) with untuned hyper-parameters, we can note that the ratio $\frac{\beta}{\alpha}$ controls the benefit of collaboration, i.e., the dependence of the "noise-term" on $M$. This is unsurprising because, as mentioned in the discussion above, making $\alpha$ small while making $\beta$ large will mimic local SGD without personalization. Further from bound (15) we can see that by making this ratio $\sqrt{M}$, we can recover the familiar noise term as in the analysis of SGD on a single machine (Stich, 2019). This might prompt one to think we have lost any benefit of collaboration. However, we note that the optimization term in the upper bound (15) is strictly better than what we get for SGD on a single machine (c.f., Section B.2).

To make this more concrete, in bound (16), we can recover the fast optimization term of pure local training and the fast noise term typical of federated learning algorithms for a specific hyper-parameter setting. Furthermore, we make no heterogeneity assumptions, meaning this upper bound is better for high enough heterogeneity than all federated learning algorithms (c.f., Table 1). This implies a regime of *"complex-enough"* problems, i.e., sufficiently ill-conditioned problems, where Algorithm 1 is strictly better than pure local training and federated learning algorithms. This is the first result of its kind, along with Bietti et al. (2022), which theoretically proves the effectiveness of personalized federated learning.

## 3.2 THE GENERAL CONVEX SETTING AND DIMINISHING CONSENSUS ERROR

The result above in the strongly convex setting is encouraging and makes us wonder what we can hope to attain in the general convex setting, i.e., when $\mu = 0$. We provide the following result in the general convex setting.

**Theorem 2** (General Convex Functions). *Assume the functions on each machine satisfy Assumptions 1 (with $\mu = 0$) and 2. For each machine $m$, we output $\hat{v}_m := \frac{1}{T} \sum_{t \in [T]} (w_t + \theta_t^m)$, where $w_t :=*

---

[2]We discuss the feasibility of this regime in Section B.2.

$\frac{1}{M} \sum_{n \in [M]} w_t^n$ *i.e., the average of all the models on that machine (* Option II *in Algorithm 1).
Then we can get the following guarantees for Algorithm 1 when* $\eta \leq \frac{1}{2L(1+\alpha)}$, $\alpha > 0$, $\beta = 1$,
$\gamma = \max\{1, M\alpha\}$ *and* $w_0 = \theta_0 = 0$,

$$\mathbb{E}\left[\frac{1}{M} \sum_{m \in [M]} F_m(\hat{v}_m) - F_m(v_m^\star)\right] \preceq \frac{B^2(1+\alpha)}{\eta \alpha T} + \frac{\eta \gamma \sigma^2}{M} + \frac{L}{T} \sum_{t=0}^{T-1} \mathbb{E}[\xi_t], \qquad (17)$$

*where* $\xi_t = \frac{1}{M} \sum_{m \in [M]} \|w_t^m - w_t\|^2$. *Alternatively, outputting* $\hat{v}_m = \frac{1}{T} \sum_{t \in [T]} (w_t^m + \theta_t^m)$
*(* Option III *in Algorithm 1), setting* $\alpha = \beta = 1$ *and using* $\eta = \min\left\{\frac{1}{4L}, \frac{\sqrt{5}B}{\sigma\sqrt{T}}\right\}$ *and* $w_0 = \theta_0 = 0$,

$$\mathbb{E}\left[\frac{1}{M} \sum_{m \in [M]} F_m(\hat{v}_m) - F_m(v_m^\star)\right] \preceq \frac{LB^2}{T} + \frac{\sigma B}{\sqrt{T}} \ . \qquad (18)$$

The blue/third term in the convergence rate presented in (17) is usually called the *consensus error*, and it frequently appears in the analyses of local update algorithms (Karimireddy et al., 2020; Woodworth et al., 2020b). It captures the cost of having local update steps, i.e., not communicating. To compare to local SGD without personalization, we re-state the convergence guarantee for local SGD (with $\beta = 1$) in terms of the consensus error (c.f., lemma 7 of Woodworth et al. (2020b)),

$$\mathbb{E}[F(\hat{w})] - \frac{1}{M} \sum_{m=1}^{M} F_m(v_m^\star) \preceq \frac{B^2}{\eta T} + \frac{\eta \sigma^2}{M} + \frac{L}{T} \sum_{t=0}^{T-1} \mathbb{E}[\xi_t] + F(v^\star) - \frac{1}{M} \sum_{m=1}^{M} F_m(v_m^\star) \qquad (19)$$

*where* $v^\star \in S^\star$, $v_m^\star \in S_m^\star$, *and* $\eta \leq \frac{1}{10L}$. This rate looks very similar to the rate presented in (17), especially when $\alpha = 1$. The main difference is the red term, which comes

from translating a non-personalized guarantee to a personalized one, as discussed in Section 2 as well as a worse noise term due to the introduction of $\gamma = \max\{1, M\alpha\}$. Even if the cost of not personalizing is small, in the local SGD analysis, there is no way[3] to control the consensus error without introducing additional heterogeneity assumptions controlling the client drift between communications rounds.

This is because between every communication round, the clients are solving their local problems, and thus, they must deviate from the optimization trajectory leading up to the optimum of the average function $F$. When we add personalization, this effect is alleviated because, with more communication, all the clients converge to stationary points (see Figure 1b). Thus,

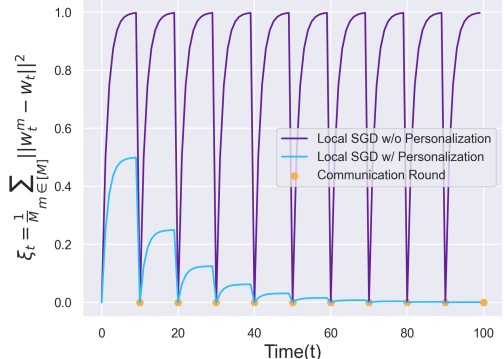

Figure 2: The evolution of consensus error with time for local SGD w/ and w/o personalization for the example presented in Proposition 1.

we expect the consensus error to go to zero with time. In the following proposition, we elucidate this gap between the average consensus of local SGD with and without personalization.

**Proposition 1** (Diminishing Consensus Error). *There exists a quadratic problem satisfying Assumptions 1, 2, and 3, such that Local SGD without personalization, i.e., Algorithm 2 with* $\beta = 1, \eta < 1$ *has averaged consensus error:*

$$\frac{1}{T} \sum_{t=0}^{T-1} \xi_t^{w/o \, pers.} = \zeta_\star^2 \cdot \left(1 + \frac{1 - (1-\eta)^{2K}}{(2-\eta)\eta K} - \frac{2(1 - (1-\eta)^K)}{\eta K}\right). \qquad (20)$$

*This error is* $\Theta(\zeta_\star^2)$ *if* $1 > \eta = \Omega(1/K)$ *for all values of* $K > 1, R \geq 1$. *On the other hand, Local SGD with personalization, i.e., Algorithm 1 with* $\beta = 1, \eta < \frac{1}{1+\alpha}$ *has averaged consensus error:*

$$\frac{1}{T} \sum_{t=0}^{T-1} \xi_t^{w/ \, pers.} \leq \frac{\zeta_\star^2}{1+\alpha} \cdot \left(1 - \frac{1 - (1-\eta(1+\alpha))^K}{\eta(1+\alpha)K}\right) \cdot \left(\frac{1 - (1-\eta(1+\alpha))^R}{\eta(1+\alpha)R}\right). \qquad (21)$$

---

[3]Consensus error is trivially upper bounded by the amount of progress SGD will make in $K$ steps on each machine, but that is usually much larger than the control provided by heterogeneity assumptions.

*The notable fact about this error upper bound is that it goes to zero with a large $R$ and can be reduced by increasing the personalization through $\alpha$.*

We also plot the consensus error as a function of time for both these algorithms and for the example (w/ $L = 1$) used in the proof of the proposition (c.f., Appendix B.6) in Figure 2. We set the step-size $\eta = \frac{1}{2L}, \beta = 1$ for local SGD (c.f., Algorithm 2), and $\alpha = 1, \beta = 1, \eta = \frac{1}{2(1+\alpha)L}$ for local SGD with Personalization, i.e., Algorithm 1. Also, for simplicity, we choose $\zeta_\star = 1$. As shown in Figure 2, consensus error goes to zero with communication and grows back to some value between communication rounds. When there is personalization, the amplitude of this value decays in every communication round, while without personalization, it stays fixed and approaches $\zeta_\star^2$. We strongly believe that this observation can be used in analyzing Algorithm 1 in the general convex setting, but currently do not have the analytical tools to provide this convergence. We conjecture that, unlike the strongly convex setting, we might need heterogeneity assumptions in the convex settings and can attain the following convergence rate for Algorithm 1.

**Conjecture 1** (General Convex Functions). *Assume the functions on each machine satisfy Assumptions 1 (with $\mu = 0$), 2 and 3. Then we can get the following guarantee for Algorithm 1 for some appropriate output $\hat{\phi}$ and hyperparameters $\eta, \alpha, \beta$,*

$$\mathbb{E}\left[\frac{1}{M}\sum_{m\in[M]} F_m(\hat{v}_m) - F_m(v_m^\star)\right] \preceq \frac{MLB^2}{T} + \frac{\sigma B}{\sqrt{MT}} + \chi_1(\sigma, K, R) + \chi_2(\zeta_\star, R),$$

*where $\lim_{K\to\infty} \chi_1(\sigma, K, R) = \lim_{R\to\infty} \chi_1(\sigma, K, R) = 0$, while $\lim_{R\to\infty} \chi_2(\zeta_\star, R) = 0$.*

The rate in the above conjecture improves over vanilla local SGD due to the strictly better optimization term $\frac{LB^2}{KR}$, which is unattainable by local SGD due to the lower bound of Patel et al. (2023). We expect this improvement will come from the ability to choose $\eta = \Theta(1/L)$ for Algorithm 1 as opposed to $\eta = \Theta(1/KL)$ for Algorithm 2, resulting in faster optimization (c.f., Equations (17) and (19)). Proving the above conjecture is an important future direction.

### 3.3 THE ROLE OF OVER-PARAMETERIZATION

So far, we have seen that personalization can alleviate the tensions in usual federated optimization in two different ways. First, the additive personalization model allows us to recover any optima $v_m^\star$ on machine $m$ by setting $\theta_m = v_m^\star - w$. This means there are infinitely many pairs of $(w, \theta_m)$ that sum up to any optimum of $F_m$, making Problem (5) an under-determined problem. As illustrated in Figure 1a, this over-parameterization (lifting the parameter space from $\mathbb{R}^d$ to $\mathbb{R}^{2d}$) expands the *"accessible solution set"*.

In the extreme case when $\cap_{m\in[M]} S_m^\star$ is an empty set, non-personalized local update algorithms can not even converge, as can be seen in Figure 1b. Between communication rounds, the iterates on each machine drift and move towards the set of optima for that machine, which repeats forever (c.f., Figure 2). With personalization, because of the personal model $\theta_t^m$, even though the machine's iterate sequences move towards the average, they do not become the same at the point of communication. This is the second benefit of over-parameterization, which stabilizes the optimization dynamics of Algorithm 1.

## 4 CONCLUSION AND DISCUSSION

In this paper, we present the first analysis of personalized local SGD in the strongly convex setting that does not require any data heterogeneity assumptions and improves over all naive baselines. There are several questions left open by our work. In terms of theoretical results, we would like to prove our Conjecture 1, thus confirming our intuition about the benefit of diminishing consensus error between the machines. Beyond this, we would also like to understand the implicit bias of Algorithm 1 and if it favors solutions concepts such as minimum norm solutions. Finally, the end goal of this research agenda is to establish a data heterogeneity notion, which characterizes the three different regimes where (i) pure local training, (ii) personalized federated learning, and (iii) vanilla federated learning are the optimal strategies, respectively.

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

# A    MISSING DETAILS FROM SECTION 2

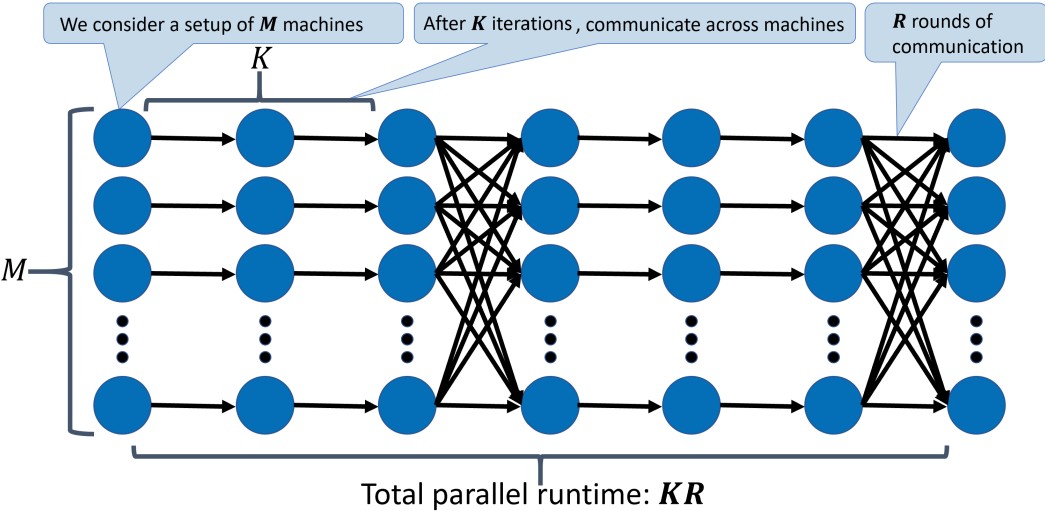

Figure 3: An intermittently communicating algorithm.

# B    MISSING DETAILS FROM SECTION 3

## B.1    LOCAL SGD WITHOUT PERSONALIZATION

---
**Algorithm 2** Local SGD
---
1: **Input:** inner step-size sequence $\eta_t$, outer step-size $\beta$, initializations $w_0$
2: Initialize $w_0^m = w_0$ on all machines $m \in [M]$
3: **for** $t \in \{0, \ldots, KR - 1\}$ **do**
4:     **for** $m \in [M]$ **in parallel do**
5:         Sample $z_t^m \sim \mathcal{D}_m$
6:         Compute gradient $\nabla_{w_t^m} f(w_t^m; z_t^m)$
7:         $w_{t+1}^m \leftarrow w_t^m - \eta_t \nabla_{w_t^m} f(w_t^m, \theta_t^m; z_t^m)$
8:         **if** $(t+1) \bmod K = 0$ **then**
9:             **Communicate** $w_{t+1}^m$ to server
10:             **Server makes update** $w_{t+1} \leftarrow w_{t+1-K} + \beta \frac{1}{M} \sum_{m' \in [M]} \left( w_{t+1}^{m'} - w_{t+1-K} \right)$
11:             **Receive** $w_{t+1}$ from server
12:             $w_{t+1}^m \leftarrow w_{t+1}$
13:         **end if**
14:     **end for**
15: **end for**
16: **Output:** $\hat{w}$
---

## B.2    COMPARISON TO PURE LOCAL TRAINING

Recall the upper bound in Equation (15). Note that the noise term (i.e., the second term) is the same as SGD on a single machine (c.f. the classic convergence result in (7) in Stich (2019)) up to numerical constants. We can show that the optimization term in Equation (15) (i.e., the first term) is better than SGD on a single machine. To do this, we need to compare the following two quantities: $e^{-R(1-e^{-2\eta\mu K})}$ and $e^{-\eta\mu KR}$. To show that the first quantity is smaller in some regimes, it is sufficient to compare the terms in the exponent: $1 - e^{-2\eta\mu K}$ and $\eta\mu K$, and show the first quantity is larger in some regimes. To see this we simply plot the functions $1 - e^{-2x}$ and $x$ in Figure

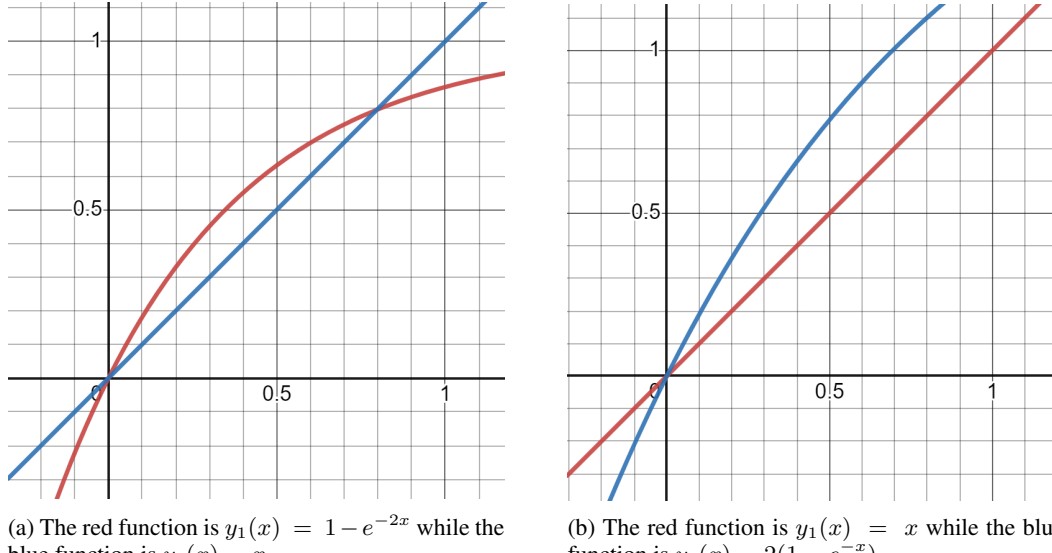

(a) The red function is $y_1(x) = 1 - e^{-2x}$ while the blue function is $y_2(x) = x$.

(b) The red function is $y_1(x) = x$ while the blue function is $y_2(x) = 2(1 - e^{-x})$.

Figure 4: Accompanying graphs to illustrate the interpretation and feasibility of the bounds in Theorem 1.

4a ($x = \eta\mu K$). We can observe that as long $\eta\mu K \leq 0.8$, i.e., $\eta \leq \frac{0.8}{\mu K}$ the convergence rate of the personalized algorithm is better than pure local training for the same $\eta$. This still raises the question of whether we can beat pure-local training when the step size is selected optimally for SGD on a single machine. This is what the bound in Equation (16) highlights. We note a step size and a regime of $\kappa$ where we can beat pure-local training with SGD (c.f., Theorem 5 in Stich (2019)). We also need to ensure that this regime of $\kappa$ is non-empty; in particular, we need to ensure that for some settings of the parameters $\frac{K}{2(1-e^{-1/4MR})} < 4MKR$. This is equivalent to ensuring $\frac{1}{4MR} < 2(1 - e^{-1/4MR})$, which in turn can be verified by setting $x = \frac{1}{4MR}$, and plotting the functions $x$ and $2(1 - e^{-x})$ in the regime $x < 1$ (because $M, R \geq 1$). We can note from Figure 4 that this regime of $\kappa$ is non-empty for every value of $M, K, R$. For instance, consider a mild parallelization with $M = 10, R = 10$ then, to see theoretical domination of personalization, we would require $\frac{K}{2(1-e^{-1/400})} \leq \kappa < 400K$ which is roughly equivalent to $201K \leq \kappa \leq 400K$. This is a fairly large range of condition numbers. We want to highlight that previous benefits of using local SGD (w/o personalization) over mini-batch SGD (and single machine SGD) have also been identified only in a regime where $\frac{\kappa}{K}$ is large (c.f., Wang et al. (2022)).

### B.3 Proof of Theorem 1

*Proof.* It would be easier to give the analysis with a slightly different notation. In particular, we denote through $w_m^{r,k}$ and $\theta_m^{r,k}$ the global and personal models on machine $m$, leading up to the $r^{th}$ communication round and after making $k$ local updates. We also denote by $w^r, \theta_m^r$ the models after the $r^{th}$ communication. In particular, we have,

$$\theta_m^r = \theta_m^{r,K} = \theta_m^{r+1,0} , \qquad\qquad \forall m \in [M] \qquad (22)$$

$$w^r = w^{r-1} + \frac{\beta}{M} \sum_{n \in [M]} \left( w_n^{r,K} - w_{r-1} \right) = w_m^{r+1,0} , \qquad \forall m \in [M] \qquad (23)$$

where recall that $\beta$ is the outer step size for the global model. The models are initialized at zero, i.e., $w^0 = w_m^{1,0} = 0$ and $\theta_m^0 = \theta_m^{1,0} = 0$ for all $m \in [M]$. We will provide the convergence for the sum of the local and global models, so we also denote the following for all $m \in [M]$, $r \in [0, R]$, and $k \in [0, K]$,

$$v_m^{r,k} := w_m^{r,k} + \theta_m^{r,k} , \qquad (24)$$
$$v_m^r := w_m^r + \theta_m^r . \qquad (25)$$

We will also slightly use the notation $f(w, \theta; z) := f(w + \theta, z)$. Now recall that the following updates happen on a machine $m$ before communication round $r$ for $k \in [0, K-1]$,

$$w_m^{r,k+1} = w_m^{r,k} - \eta \nabla f(v_m^{r,k}; z_m^{r,k}) \ , \tag{26}$$

$$\theta_m^{r,k+1} = \theta_m^{r,k} - \alpha \eta \nabla f(v_m^{r,k}; z_m^{r,k}) \ , \tag{27}$$

where $z_m^{r,k}$ is the sample drawn for the update. Adding and subtracting the exact gradient, we get,

$$w_m^{r,k+1} = w_m^{r,k} - \eta \nabla F_m(v_m^{r,k}) + \eta \left( \nabla F_m(v_m^{r,k}) - \nabla f(v_m^{r,k}; z_m^{r,k}) \right) \ , \tag{28}$$

$$\theta_m^{r,k+1} = \theta_m^{r,k} - \alpha \eta \nabla F_m(v_m^{r,k}) + \alpha \eta \left( \nabla F_m(v_m^{r,k}) - \nabla f(v_m^{r,k}; z_m^{r,k}) \right) \ . \tag{29}$$

Let us denote the error due to stochasticity by $\delta_m^{r,k}$ then we get that,

$$w_m^{r,k+1} = w_m^{r,k} - \eta \nabla F_m(v_m^{r,k}) + \eta \delta_m^{r,k} \ , \tag{30}$$

$$\theta_m^{r,k+1} = \theta_m^{r,k} - \alpha \eta \nabla F_m(v_m^{r,k}) + \alpha \eta \delta_m^{r,k} \ . \tag{31}$$

Applying the mean-value theorem for the above updates, we get that,

$$w_m^{r,k+1} = w_m^{r,k} - \eta A_m(v_m^{r,k}, v_m^\star)(v_m^{r,k} - v_m^\star) + \eta \delta_m^{r,k} \ , \tag{32}$$

$$\theta_m^{r,k+1} = \theta_m^{r,k} - \alpha \eta A_m(v_m^{r,k}, v_m^\star)(v_m^{r,k} - v_m^\star) + \alpha \eta \delta_m^{r,k} \ , \tag{33}$$

where $A_m(.,.)$ is an operator function such that for every $v, v'$, $\mu \cdot I \preceq A_m(v, v') \preceq L \cdot I$. For simplicity, we can denote the updates as follows,

$$w_m^{r,k+1} = w_m^{r,k} - \eta A_m^{r,k}(v_m^{r,k} - v_m^\star) + \eta \delta_m^{r,k} \ , \tag{34}$$

$$\theta_m^{r,k+1} = \theta_m^{r,k} - \alpha \eta \nabla A_m^{r,k}(v_m^{r,k} - v_m^\star) + \alpha \eta \delta_m^{r,k} \ , \tag{35}$$

where we must not forget that $A_m^{r,k}$ is a random matrix. By adding equations 34, 35 and by subtracting equation 35 from $\alpha$ times equation 34, we can get the following updates for $k \in [0, K-1]$,

$$v_m^{r,k+1} = v_m^{r,k} - \eta(1+\alpha)A_m^{r,k}(v_m^{r,k} - v_m^\star) + \eta(1+\alpha)\delta_m^{r,k} \ , \tag{36}$$

$$\alpha w_m^{r,k+1} - \theta_m^{r,k+1} = \alpha w_m^{r,k} - \theta_m^{r,k} \ . \tag{37}$$

Now we subtract the optima appropriately above and instead, consider the iterates centered around their fixed points, i.e., $\tilde{v}_m^{\cdots} := v_m^{\cdots} - v_m^\star$, $\tilde{\theta}_m^{\cdots} := \theta_m^{\cdots} - \theta_m^\star$, and $\tilde{w}_m^{\cdots} := w_m^{\cdots} - w^\star$,

$$\tilde{v}_m^{r,k+1} = \left( I - \eta(1+\alpha)A_m^{r,k} \right) \tilde{v}_m^{r,k} + \eta(1+\alpha)\delta_m^{r,k} \ , \tag{38}$$

$$\alpha \tilde{w}_m^{r,k+1} - \tilde{\theta}_m^{r,k+1} = \alpha \tilde{w}_m^{r,k} - \tilde{\theta}_m^{r,k} \ . \tag{39}$$

Unrolling these two recursions and setting $k = K-1$ gives us,

$$\tilde{v}_m^{r,K} = \prod_{k=0}^{K-1} \left( I - \eta(1+\alpha)A_m^{r,k} \right) \tilde{v}_m^{r-1} + \eta(1+\alpha) \sum_{k=0}^{K-1} \prod_{j=K-k}^{K-1} (I - \eta(1+\alpha)A_m^{r,j})\delta_m^{r,K-1-k} \ , \tag{40}$$

$$=: B_m^r \tilde{v}_m^{r-1} + \eta(1+\alpha)\delta_m^r \ , \tag{41}$$

$$\alpha \tilde{w}_m^{r,K} - \tilde{\theta}_m^{r,K} = \alpha \tilde{w}^{r-1} - \tilde{\theta}_m^{r-1} \ , \tag{42}$$

where we denote $\prod_{j=K}^{K-1}(I - \eta(1+\alpha)A_m^{r,j}) = I$. Adding the equations 40, 42 and then subtracting the equation 42 from alpha times equation 40, we get,

$$(1+\alpha)\tilde{w}_m^{r,K} = (\alpha I + B_m^r) \tilde{w}^{r-1} - (I - B_m^r) \tilde{\theta}_m^{r-1} + \eta(1+\alpha)\delta_m^r \ , \tag{43}$$

$$(1+\alpha)\tilde{\theta}_m^{r,K} = -\alpha (I - B_m^r) \tilde{w}^{r-1} + (I + \alpha B_m^r) \tilde{\theta}_m^{r-1} + \alpha \eta(1+\alpha)\delta_m^r \ . \tag{44}$$

Re-arranging these gives,

$$\tilde{w}_m^{r,K} = \tilde{w}^{r-1} - \frac{I - B_m^r}{1+\alpha}\tilde{v}^{r-1} + \eta \delta_m^r \ , \tag{45}$$

$$\tilde{\theta}_m^r = \tilde{\theta}_m^{r-1} - \frac{\alpha (I - B_m^r)}{1+\alpha}\tilde{v}^{r-1} + \alpha \eta \delta_m^r \ , \tag{46}$$

using the first equation, we can compute $w^r$ as follows,

$$\tilde{w}^r = \tilde{w}^{r-1} + \frac{\beta}{M} \sum_{m \in [M]} \left( \tilde{w}_m^{r,K} - \tilde{w}_{r-1} \right) \quad, \tag{47}$$

$$= \tilde{w}^{r-1} + \frac{-\beta}{M(1+\alpha)} \sum_{m \in [M]} (I - B_m^r) \tilde{v}^{r-1} + \frac{\eta\beta}{M} \sum_{m \in [M]} \delta_m^r \quad. \tag{48}$$

Thus we are finally ready to write a recursion for $\tilde{v}_m^r = \tilde{w}^r + \tilde{\theta}_m^r$ by adding equations [47] and [48],

$$\tilde{v}_m^r = \tilde{v}_m^{r-1} - \frac{\beta}{M(1+\alpha)} \sum_{n \in [M]} (I - B_n^r) \tilde{v}_n^{r-1} - \frac{\alpha(I - B_m^r)}{1+\alpha} \tilde{v}_m^r + \frac{\eta\beta}{M} \sum_{m \in [M]} \delta_m^r + \alpha\eta\delta_m^r \quad. \tag{49}$$

Thus if we denote the vector $\tilde{\phi}^r := (\tilde{v}_1^r, \ldots, \tilde{v}_M^r)$ we can write the following recursion,

$$\tilde{\phi}^r = \tilde{\phi}^{r-1} - \frac{1}{1+\alpha} \begin{bmatrix} \left(\alpha + \frac{\beta}{M}\right)(I - B_1^r) & \cdots & \frac{\beta}{M}(I - B_M^r) \\ \vdots & \ddots & \vdots \\ \frac{\beta}{M}(I - B_1^r) & \cdots & \left(\alpha + \frac{\beta}{M}\right)(I - B_M^r) \end{bmatrix} \tilde{\phi}^{r-1} + \eta \begin{bmatrix} \frac{\beta}{M}\sum_{m \in [M]} \delta_m^r + \alpha\delta_1^r \\ \vdots \\ \frac{\beta}{M}\sum_{m \in [M]} \delta_m^r + \alpha\delta_M^r \end{bmatrix}, \tag{50}$$

$$=: \left( I - \frac{G^r}{1+\alpha} \right) \tilde{\phi}^{r-1} + \eta\delta^r \quad. \tag{51}$$

Taking the norm and squaring on both sides, following by taking expectations we get,

$$\mathbb{E} \left\| \tilde{\phi}^r \right\|_2^2 = \mathbb{E} \left\| \left( I - \frac{G^r}{1+\alpha} \right) \tilde{\phi}^{r-1} + \eta\delta^r \right\|_2^2 \quad, \tag{52}$$

$$= \mathbb{E} \left\| \left( I - \frac{G^r}{1+\alpha} \right) \tilde{\phi}^{r-1} \right\|_2^2 + \eta^2 \mathbb{E} \left\| \delta^r \right\|_2^2 \quad, \tag{53}$$

$$\leq \mathbb{E} \left\| I - \frac{G^r}{1+\alpha} \right\|_2^2 \left\| \phi^{r-1} \right\|_2^2 + \eta^2 \mathbb{E} \left\| \delta^r \right\|_2^2 \quad, \tag{54}$$

the second equality follows because we have i.i.d. samples on each machine at each time step. Note that when $0 < \eta < \frac{1}{L(1+\alpha)}$ then $0 \prec B_m^r \prec I$, which ensures that $I - B_m^r \succ 0$. This makes $G^r$ a block matrix of positive definite matrices, and in particular, it is a very structured matrix that can be decomposed into a block diagonal matrix plus a positive semi-definite matrix with the same block rows by pulling away the terms with $\beta$ and $\alpha$. This decomposition makes it possible to compute its spectrum exactly. As long as $\beta > 0$, one can quickly verify the following lower bound on the smallest eigenvalue of $G^r$,

$$\lambda_{min}(G^r) \geq \alpha \min_{m \in [M]} \lambda_{min}(I - B_m^r), \tag{55}$$

$$\geq \alpha \left( 1 - \max_{m \in [M]} \lambda_{max}(B_m^r) \right) \quad, \tag{56}$$

$$= \alpha \left( 1 - \max_{m \in [M]} \lambda_{max} \left( \prod_{k=0}^{K-1} (I - \eta(1+\alpha)A_m^{r,k}) \right) \right) \quad, \tag{57}$$

$$\geq \alpha \left( 1 - \prod_{k=0}^{K-1} \max_{m \in [M]} \lambda_{max} \left( I - \eta(1+\alpha)A_m^{r,k} \right) \right) \quad, \tag{58}$$

$$= \alpha \left( 1 - \prod_{k=0}^{K-1} \max_{m \in [M]} \left( 1 - \eta(1+\alpha)\lambda_{min}(A_m^{r,k}) \right) \right) \quad, \tag{59}$$

$$= \alpha \left( 1 - \prod_{k=0}^{K-1} \left( 1 - \eta(1+\alpha) \min_{m \in [M]} \lambda_{min}(A_m^{r,k}) \right) \right) \quad, \tag{60}$$

$$\geq \alpha \left( 1 - \prod_{k=0}^{K-1} (1 - \eta(1+\alpha)\mu) \right) , \tag{61}$$

$$= \alpha \left( 1 - (1 - \eta(1+\alpha)\mu)^K \right) . \tag{62}$$

With this, we can note the following in equation 54,

$$\mathbb{E} \left\| \tilde{\phi}^r \right\|_2^2 \leq \mathbb{E} \left\| I - \frac{G^r}{1+\alpha} \right\|_2^2 \left\| \phi^{r-1} \right\|_2^2 + \eta^2 \mathbb{E} \left\| \delta^r \right\|_2^2 , \tag{63}$$

$$= \mathbb{E}\lambda_{max} \left( I - \frac{G^r}{1+\alpha} \right)^2 \left\| \phi^{r-1} \right\|_2^2 + \eta^2 \mathbb{E} \left\| \delta^r \right\|_2^2 , \tag{64}$$

$$\leq \left( 1 - \frac{\alpha}{1+\alpha} \left( 1 - (1 - \eta(1+\alpha)\mu)^K \right) \right)^2 \mathbb{E} \left\| \phi^{r-1} \right\|_2^2 + \eta^2 \mathbb{E} \left\| \delta^r \right\|_2^2 . \tag{65}$$

Let us first consider the exact setting when the stochastic terms are zero; in that case, we can simplify bound 65 for $r = R$ to,

$$\mathbb{E} \left\| \tilde{\phi}^R \right\|_2^2 \leq \left( 1 - \frac{\alpha}{1+\alpha} \left( 1 - (1 - \eta(1+\alpha)\mu)^K \right) \right)^2 \mathbb{E} \left\| \phi^{R-1} \right\|_2^2 , \tag{66}$$

$$\leq \left( 1 - \frac{\alpha}{1+\alpha} \left( 1 - (1 - \eta(1+\alpha)\mu)^K \right) \right)^{2R} \mathbb{E} \left\| \phi^0 \right\|_2^2 , \tag{67}$$

$$\leq e^{-\frac{2\alpha R}{1+\alpha} \left( 1 - e^{-\eta(1+\alpha)\mu K} \right)} , \tag{68}$$

$$\leq e^{-\frac{2\alpha R}{1+\alpha} \left( 1 - e^{-\frac{K}{2\kappa}} \right)} , \tag{69}$$

where we just set the step-size to $\eta = \frac{1}{2L(1+\alpha)}$. This finishes the proof for the deterministic setting.

Now consider the stochastic setting. Let us carefully upper-bound the second term in bound 65. To do so, note that the noise of sampling data is independent across machines and time. This gives us the following decomposition,

$$\mathbb{E} \left\| \delta^r \right\|_2^2 = \sum_{m \in [M]} \mathbb{E} \left\| \frac{\beta}{M} \sum_{n \in [M]} \delta_n^r + \alpha \delta_m^r \right\|_2^2 , \tag{70}$$

$$= \sum_{m \in [M]} \mathbb{E} \left\| \frac{\beta}{M} \sum_{n \neq m} \delta_n^r + \left( \alpha + \frac{\beta}{M} \right) \delta_m^r \right\|_2^2 , \tag{71}$$

$$= \sum_{m \in [M]} \left( \left( \alpha + \frac{\beta}{M} \right)^2 \mathbb{E} \left\| \delta_m^r \right\|_2^2 + \frac{\beta^2}{M^2} \sum_{n \neq m} \mathbb{E} \left\| \delta_n^r \right\|_2^2 \right) , \tag{72}$$

$$= \left( \left( \alpha + \frac{\beta}{M} \right)^2 + (M-1)\frac{\beta^2}{M^2} \right) \sum_{m \in [M]} \mathbb{E} \left\| \delta_m^r \right\|_2^2 , \tag{73}$$

$$\leq \left( 2\alpha^2 + \frac{\beta^2}{M^2} + \frac{\beta^2}{M} \right) \sum_{m \in [M]} \mathbb{E} \left\| \delta_m^r \right\|_2^2 , \tag{74}$$

$$\leq \left( 2\alpha^2 + \frac{2\beta^2}{M} \right) \sum_{m \in [M]} \mathbb{E} \left\| \delta_m^r \right\|_2^2 , \tag{75}$$

Further continuing after noting that the stochastic noise is independent and mean-zero across local steps, we get that,

$$\mathbb{E} \left\| \delta^r \right\|_2^2 \leq 2 \left( \alpha^2 + \frac{\beta^2}{M} \right) \sum_{m \in [M]} \sum_{k=0}^{K-1} \mathbb{E} \left\| \prod_{j=K-k}^{K-1} (I - \eta(1+\alpha)A_m^{r,j}) \delta_m^{r,K-1-k} \right\|_2^2 , \tag{76}$$

$$\leq 2 \left( \alpha^2 + \frac{\beta^2}{M} \right) \sum_{k=0}^{K-1} \sum_{m \in [M]} \mathbb{E} \prod_{j=K-k}^{K-1} \left\| (I - \eta(1+\alpha)A_m^{r,j}) \right\|_2^2 \left\| \delta_m^{r,K-1-k} \right\|_2^2 \quad , \tag{77}$$

$$\leq 2 \left( \alpha^2 + \frac{\beta^2}{M} \right) M \sum_{k=0}^{K-1} (1 - \eta(1+\alpha)\mu)^{2k} \mathbb{E} \left\| \delta_m^{r,K-1-k} \right\|_2^2 \quad , \tag{78}$$

$$\leq 2 \left( \alpha^2 M + \beta^2 \right) \sigma^2 \sum_{k=0}^{K-1} (1 - \eta(1+\alpha)\mu)^{2k} \quad , \tag{79}$$

$$= 2 \left( \alpha^2 M + \beta^2 \right) \sigma^2 \frac{1 - (1 - \eta(1+\alpha)\mu)^{2K}}{1 - (1 - \eta(1+\alpha)\mu)^2} \quad , \tag{80}$$

$$\leq 2 \left( \alpha^2 M + \beta^2 \right) \sigma^2 \frac{1 - (1 - \eta(1+\alpha)\mu)^{2K}}{\eta(1+\alpha)\mu} \quad , \tag{81}$$

where in the last inequality we use that $\eta \leq \frac{1}{(1+\alpha)\mu}$. Plugging bound 81 in bound 65, we get,

$$\mathbb{E} \left\| \tilde{\phi}^r \right\|_2^2 \leq \left( 1 - \frac{\alpha}{1+\alpha} \left( 1 - (1 - \eta(1+\alpha)\mu)^K \right) \right)^2 \mathbb{E} \left\| \phi^{r-1} \right\|_2^2 + \frac{2\eta \left( \alpha^2 M + \beta^2 \right) \sigma^2}{(1+\alpha)\mu} \left( 1 - (1 - \eta(1+\alpha)\mu)^{2K} \right) \quad , \tag{82}$$

Setting $r = R$ and unrolling this recursion and averaging we get, we get,

$$\frac{1}{M} \mathbb{E} \left\| v_m^R - v_m^\star \right\|_2^2 \leq \left( 1 - \frac{\alpha}{1+\alpha} \left( 1 - (1 - \eta(1+\alpha)\mu)^K \right) \right)^{2R} \frac{1}{M} \sum_{m \in [M]} \left\| v_m^\star \right\|_2^2 \tag{83}$$

$$+ \frac{2\eta \left( \alpha^2 M + \beta^2 \right) \sigma^2}{M(1+\alpha)\mu} \cdot \frac{1 - (1 - \eta(1+\alpha)\mu)^{2K}}{1 - \left( 1 - \frac{\alpha}{1+\alpha} \left( 1 - (1 - \eta(1+\alpha)\mu)^K \right) \right)^2} \quad , \tag{84}$$

$$\leq \left( 1 - \frac{\alpha}{1+\alpha} \left( 1 - (1 - \eta(1+\alpha)\mu)^K \right) \right)^{2R} B^2 \tag{85}$$

$$+ \frac{2\eta \left( \alpha^2 M + \beta^2 \right) \sigma^2}{M\alpha\mu} \cdot \frac{1 - (1 - \eta(1+\alpha)\mu)^{2K}}{1 - (1 - \eta(1+\alpha)\mu)^K} \quad , \tag{86}$$

$$\leq \left( 1 - \frac{\alpha}{1+\alpha} \left( 1 - e^{-\eta(1+\alpha)\mu K} \right) \right)^{2R} B^2 + \frac{4\eta \left( \alpha^2 M + \beta^2 \right) \sigma^2}{M\alpha\mu} \quad , \tag{87}$$

$$\leq e^{-\frac{2\alpha R}{1+\alpha} \left( 1 - e^{-\eta(1+\alpha)\mu K} \right)} B^2 + \frac{4\eta \left( \alpha^2 M + \beta^2 \right) \sigma^2}{M\alpha\mu} \quad , \tag{88}$$

$$= e^{-\frac{2\alpha R}{1+\alpha} \left( 1 - e^{-\eta(1+\alpha)\mu K} \right)} B^2 + \frac{4\eta\sigma^2}{\mu} \left( \alpha + \frac{\beta^2}{M\alpha} \right) \quad , \tag{89}$$

$$= e^{-\frac{2\beta R}{\sqrt{M}+\beta} \left( 1 - e^{-\eta(1+\beta/\sqrt{M})\mu K} \right)} B^2 + \frac{8\eta\beta\sigma^2}{\sqrt{M}\mu} \quad , \tag{90}$$

$$= e^{-R \left( 1 - e^{-2\eta\mu K} \right)} B^2 + \frac{8\eta\sigma^2}{\mu} \quad , \tag{91}$$

where we pick $\alpha = \frac{\beta}{\sqrt{M}}$ and $\beta = \sqrt{M}$ and $\eta$ is constrained by $\eta < \frac{1}{2L}$. Now let us pick $\eta = \frac{1}{8\mu MKR}$, this is possible only when $\kappa < 4MKR$, then we get the following rate,

$$\frac{1}{M} \mathbb{E} \left\| v_m^R - v_m^\star \right\|_2^2 \leq e^{-R \left( 1 - e^{-\frac{1}{4MR}} \right)} B^2 + \frac{\sigma^2}{\mu^2 MKR} \quad . \tag{92}$$

Now we want the first term to match the bias term of the non-collaborative baseline asymptotically. In particular, note that we can show the following when $\kappa \geq \frac{K}{2(1 - e^{-\frac{1}{4MR}})}$,

$$\kappa \geq \frac{K}{2(1 - e^{-\frac{1}{4MR}})} \quad , \tag{93}$$

$$\equiv (1 - e^{-\frac{1}{4MR}}) \geq \frac{K}{2\kappa} \ , \tag{94}$$

$$\equiv R(1 - e^{-\frac{1}{4MR}}) \geq \frac{KR}{2\kappa} \ , \tag{95}$$

$$\equiv e^{-R\left(1 - e^{-\frac{1}{4MR}}\right)} B^2 \leq e^{-\frac{KR}{2\kappa}} B^2 \ , \tag{96}$$

which is exactly the bias term of the non-collaborative baseline. Now all we need to check is if the two conditions on the condition number $\kappa$ can be satisfied simultaneously. In particular, we want to satisfy the following:

$$\frac{K}{2(1 - e^{-\frac{1}{4MR}})} < 4MKR \ , \tag{97}$$

$$\equiv \frac{1}{8MR} < 1 - e^{-\frac{1}{4MR}} \ , \tag{98}$$

$$\equiv \frac{y}{2} < 1 - e^{-y}, \ y = \frac{1}{4MR} \ , \tag{99}$$

the above condition is always true when $\frac{1}{4MR} \leq 1$, which is always true as $M, R \geq 1$. To see this, note the functions plotted in Figure 4. Furthermore, note that $\kappa = \frac{L}{\mu} \geq 1$. Thus the assumption lower bounding $\kappa$ is binding only when,

$$1 \leq \frac{K}{2(1 - e^{-1/4MR})} \ , \tag{100}$$

$$\equiv 2(1 - e^{-1/4MR}) \leq K \ . \tag{101}$$

Note that since $MR > 1$, the left-hand term above is always smaller than 1, which means the lower bound on $\kappa$ is always binding. This finishes the proof. $\qquad\square$

### B.4 PROOF OF THEOREM 2

*Proof.* Throughout the proof, we will assume $\mathcal{W} = \mathbb{R}^d$. We will first define the following lifted vectors that will be useful for our analysis:

$$w_t := \frac{1}{M} \sum_{m \in [M]} w_t^m, \ \psi_t^m = \begin{bmatrix} w_t^m \\ \theta_t^1 \\ \vdots \\ \theta_t^M \end{bmatrix}, \ \psi_t = \frac{1}{M} \sum_{m \in [M]} \psi_t^m = \begin{bmatrix} w_t \\ \theta_t^1 \\ \vdots \\ \theta_t^M \end{bmatrix}.$$

We also define the following stochastic gradient vector for all $m \in [M]$ by lifting the dimension of the stochastic gradient of $F_m$,

$$i_t^m = \begin{bmatrix} \nabla_w f_m(w_t^m + \theta_t^m; z_t^m) \\ 0 \\ \vdots \\ \nabla_\theta f(w_t^m + \theta_t^m; z_t^m) \\ \vdots \\ 0 \end{bmatrix}$$

We will also define $i_t := \frac{1}{M} \sum_{m \in [M]} i_t^m$ as well as its conditional expectation w.r.t. the filtration $\mathcal{H}_t = \sigma\left(w_t^1, \ldots, w_t^M, \theta_t^1, \ldots, \theta_t^M\right)$,

$$g_t^m := \mathbb{E}[i_t^m | \mathcal{H}_t] = \nabla_\psi \hat{F}_m(w_t^m, \theta_t^m) = \begin{bmatrix} \nabla_w F_m(w_t^m + \theta_t^m) \\ 0 \\ \vdots \\ \nabla_\theta F_m(w_t^m + \theta_t^m) \\ \vdots \\ 0 \end{bmatrix}$$

The following two gradients will also be useful,

$$
g_t = \frac{1}{M} \sum_{m \in [M]} g_t^m = \begin{bmatrix} \frac{1}{M} \sum_{m \in [M]} \nabla_w F_m(w_t^m + \theta_t^m) \\ \frac{1}{M} \nabla_\theta F_1(w_t^1 + \theta_t^1) \\ \vdots \\ \frac{1}{M} \nabla_\theta F_M(w_t^M + \theta_t^M) \end{bmatrix}, \quad h_t = \begin{bmatrix} \frac{1}{M} \sum_{m \in [M]} \nabla_w F_m(w_t + \theta_t^m) \\ \frac{1}{M} \nabla_\theta F_1(w_t + \theta_t^1) \\ \vdots \\ \frac{1}{M} \nabla_\theta F_M(w_t + \theta_t^M) \end{bmatrix},
$$

where we can note that $g_t \neq h_t$ unless $(t) \bmod K = 0$. We will also consider an arbitrary parameterization of the optima of different machines as,

$$
\psi^\star = \begin{bmatrix} w^\star \\ \theta_1^\star \\ \vdots \\ \theta_M^\star \end{bmatrix} \in \mathcal{W}^{M+1}, \; w^\star + \theta_m^\star \in \arg\min_{v \in \mathcal{W}} F_m(v), \; F^\star := \hat{F}(\psi^\star).
$$

Since this parameterization is arbitrary, we can choose the minimum-norm parameterization (in any norm). We will use this fact later. Finally, we define,

$$
H = \begin{bmatrix} I_d & 0 \\ 0 & \frac{1}{M\alpha} I_{Md} \end{bmatrix}, \text{ and } H^{-1} = \begin{bmatrix} I_d & 0 \\ 0 & M\alpha I_{Md} \end{bmatrix}.
$$

Note that if $\alpha = 0$, we will reduce to the non-personalized setting, and we can let $H = H^{-1} = I_d$ and not lift the dimension of the parameter space, i.e., $\phi = w \in \mathcal{W}$. With this disclaimer, we can write for $\alpha > 0$,

$$
\psi_{t+1} = \begin{bmatrix} \frac{1}{M} \sum_{m \in [M]} w_t^m - \eta_t \nabla_w F_m(w_t^m + \theta_t^m) \\ \theta_t^1 - \alpha \eta_t \nabla_\theta F_1(w_t^1 + \theta_t^1) \\ \vdots \\ \theta_t^M - \alpha \eta_t \nabla_\theta F_M(w_t^M + \theta_t^M) \end{bmatrix} + \begin{bmatrix} \frac{\eta_t}{M} \sum_{m \in [M]} (\nabla_w F_m(w_t^m + \theta_t^m) - \nabla_w f(w_t^m + \theta_t^m; z_t^m)) \\ \alpha \eta_t (\nabla_\theta F_1(w_t^1 + \theta_t^1) - \nabla_\theta f(w_t^1 + \theta_t^1; z_t^1)) \\ \vdots \\ \alpha \eta_t (\nabla_\theta F_M(w_t^M + \theta_t^M) - \nabla_\theta f(w_t^M + \theta_t^M; z_t^M)) \end{bmatrix},
$$

$$
= \psi_t - \eta_t \begin{bmatrix} \frac{1}{M} \sum_{m \in [M]} \nabla_w F_m(w_t^m + \theta_t^m) \\ \alpha \nabla_\theta F_1(w_t^1 + \theta_t^1) \\ \vdots \\ \alpha \nabla_\theta F_M(w_t^M + \theta_t^M) \end{bmatrix} + \eta_t \begin{bmatrix} \frac{1}{M} \sum_{m \in [M]} (\nabla_w F_m(w_t^m + \theta_t^m) - \nabla_w f(w_t^m + \theta_t^m; z_t^m)) \\ \alpha (\nabla_\theta F_1(w_t^1 + \theta_t^1) - \nabla_\theta f(w_t^1 + \theta_t^1; z_t^1)) \\ \vdots \\ \alpha (\nabla_\theta F_M(w_t^M + \theta_t^M) - \nabla_\theta f(w_t^M + \theta_t^M; z_t^M)) \end{bmatrix},
$$

$$
= \psi_t - \eta_t H^{-1} g_t + \eta_t H^{-1} (g_t - i_t).
$$

First let us bound the expected norm of the last term as follows,

$$
\mathbb{E} \left[ \left\| H^{-1} (g_t - i_t) \right\|_H^2 \mid \mathcal{H}_t \right] = \mathbb{E} \left[ \left\| g_t - i_t \right\|_{H^{-1}}^2 \mid \mathcal{H}_t \right],
$$

$$
= \mathbb{E} \left[ \left\| \frac{1}{M} \sum_{m \in [M]} \nabla_w f(w_t^m + \theta_t^m; z_t^m) - \nabla_w F_m(w_t^m + \theta_t^m) \right\|^2 \right.
$$

$$
\left. + \alpha M \sum_{m \in [M]} \frac{1}{M^2} \left\| \nabla_\theta f(w_t^m + \theta_t^m; z_t^m) - \nabla_\theta F_m(w_t^m + \theta_t^m) \right\|^2 \mid \mathcal{H}_t \right],
$$

$$
= \frac{1}{M^2} \sum_{m \in [M]} \mathbb{E} \left[ \left\| \nabla_w f(w_t^m + \theta_t^m; z_t^m) - \nabla_w F_m(w_t^m + \theta_t^m) \right\|^2 \right]
$$

$$
+ \frac{\alpha}{M} \sum_{m \in [M]} \mathbb{E} \left[ \left\| \nabla_\theta f(w_t^m + \theta_t^m; z_t^m) - \nabla_\theta F_m(w_t^m + \theta_t^m) \right\|^2 \mid \mathcal{H}_t \right],
$$

$$
\leq \frac{\sigma^2}{M} + \alpha \sigma^2.
$$

Defining $\gamma = \max\{1, \alpha M\}$ we can re-write this as,

$$
\mathbb{E} \left[ \left\| g_t - i_t \right\|_{H^{-1}}^2 \mid \mathcal{H}_t \right] \leq 2\gamma \frac{\sigma^2}{M}. \tag{102}
$$

We will use this bound now. First subtracting $\psi^\star$ from $\psi_{t+1}$ and taking the norm, we can write the following recurrence,

$$\mathbb{E}\left[\left\|\psi_{t+1} - \psi^\star\right\|_H^2 |\mathcal{H}_t\right] = \mathbb{E}\left[\left\|\psi_t - \eta_t H^{-1} g_t + \eta_t H^{-1}(g_t - i_t) - \psi^\star\right\|_H^2 |\mathcal{H}_t\right],$$

$$= \|\psi_t - \psi^\star\|_H^2 + \eta_t^2\|g_t\|_{H^{-1}}^2 - 2\eta_t\langle g_t, \psi_t - \psi^\star\rangle + \eta_t^2\mathbb{E}\left[\left\|\frac{1}{M}\sum_{m\in[M]}(g_t^m - i_t^m)\right\|_{H^{-1}}^2 |\mathcal{H}_t\right],$$

$$\leq \|\psi_t - \psi^\star\|_H^2 + 2\eta_t^2\|g_t - h_t\|_{H^{-1}}^2 + 2\eta_t^2\|h_t\|_{H^{-1}}^2 - 2\eta_t\frac{1}{M}\sum_{m\in[M]}\langle g_t^m, \psi_t - \psi^\star\rangle + \frac{2\eta_t^2\gamma\sigma^2}{M}.$$
$$(103)$$

Now, we can upper-bound the blue and red terms separately. First, let's bound the red term as follows,

$$\|h_t\|_{H^{-1}}^2 = \left\|h_t - \nabla_\psi\hat{F}(\psi^\star)\right\|_{H^{-1}}^2,$$

$$= \left\|\frac{1}{M}\sum_{m\in[M]}\nabla_w\hat{F}_m(w_t, \theta_t^m) - \nabla_w\hat{F}_m(w^\star, \theta_m^\star)\right\|^2 + M\alpha\sum_{m\in[M]}\frac{1}{M^2}\left\|\nabla_\theta\hat{F}_m(w_t, \theta_t^m) - \nabla_\theta\hat{F}_m(w^\star, \theta_m^\star)\right\|^2,$$

$$\leq \frac{1}{M}\sum_{m\in[M]}\left\|\nabla_w\hat{F}_m(w_t, \theta_t^m) - \nabla_w\hat{F}_m(w^\star, \theta_m^\star)\right\| + \frac{\alpha}{M}\sum_{m\in[M]}\left\|\nabla_\theta\hat{F}_m(w_t, \theta_t^m) - \nabla_\theta\hat{F}_m(w^\star, \theta_m^\star)\right\|^2,$$

$$\leq \frac{1+\alpha}{M}\sum_{m\in[M]}\left\|\nabla_\psi\hat{F}_m(w_t, \theta_t^m) - \nabla_\psi\hat{F}_m(w^\star, \theta_m^\star)\right\|^2,$$

$$\leq \frac{1+\alpha}{M}\sum_{m\in[M]}\left\|\nabla_\psi\hat{F}_m(w_t, \theta_t^m) - \nabla_\psi\hat{F}_m(w^\star, \theta_m^\star)\right\|^2,$$

$$\leq \frac{1+\alpha}{M}\sum_{m\in[M]}2L\left(\hat{F}_m(w_t^m, \theta_t^m) - \hat{F}_m(w^\star, \theta_m^\star) + \left\langle\nabla_\psi\hat{F}_m(w^\star, \theta_m^\star), \psi_t - \psi^\star\right\rangle\right),$$

$$= 2L(1+\alpha)\cdot\left(\hat{F}(\psi_t) - \hat{F}(\psi^\star) + \left\langle\nabla_\psi\hat{F}(\psi^\star), \psi_t - \psi^\star\right\rangle\right),$$

$$= 2L(1+\alpha)\cdot\left(\hat{F}(\psi_t) - \hat{F}(\psi^\star)\right),$$

$$= 2L(1+\alpha)\cdot\left(\hat{F}(\psi_t) - \hat{F}(\psi^\star)\right).$$

Next, we will bound the blue term as follows,

$$\|g_t - h_t\|_{H^{-1}}^2 = \left\|\frac{1}{M}\sum_{m\in[M]}\nabla_w\hat{F}_m(w_t^m, \theta_t^m) - \nabla_w\hat{F}_m(w_t, \theta_t^m)\right\|^2 + \frac{\alpha}{M}\sum_{m\in[M]}\left\|\nabla_\theta\hat{F}_m(w_t^m, \theta_t^m) - \nabla_\theta\hat{F}_m(w_t, \theta_t^m)\right\|^2,$$

$$\leq \frac{L^2}{M}\sum_{m\in[M]}\|w_t^m - w_t\|^2 + \frac{L^2\alpha}{M}\sum_{m\in[M]}\|w_t^m - w_t\|^2,$$

$$\leq \frac{L^2(1+\alpha)}{M}\sum_{m\in[M]}\|w_t^m - w_t\|^2,$$

$$\leq L^2(1+\alpha)\xi_t,$$

where we define $\xi_t := \frac{1}{M}\sum_{m\in[M]}\|w_t^m - w_t\|^2$, i.e., the consensus error (Karimireddy et al., 2020; Woodworth et al., 2020b) of the shared model. Putting back the upper bounds on the red and blue terms in (103) above, and taking full expectation we get,

$$\mathbb{E}\|\psi_{t+1} - \psi^\star\|_H^2 \leq \mathbb{E}\|\psi_t - \psi^\star\|_H^2 + 2\eta_t^2(1+\alpha)L^2\mathbb{E}\xi_t + 2\eta_t^2 L(1+\alpha)\cdot\mathbb{E}\left[F(\psi_t) - F(\psi^\star)\right]$$

$$- \frac{2\eta_t}{M} \sum_{m \in [M]} \mathbb{E} \langle g_t^m, \psi_t - \psi^\star \rangle + \frac{2\eta_t^2 \gamma \sigma^2}{M},$$

$$= \mathbb{E} \|\psi_t - \psi^\star\|_H^2 + 2\eta_t^2 L^2 (1 + \alpha) \mathbb{E} \xi_t + 2\eta_t^2 L(1 + \alpha) \cdot \mathbb{E} \left[ F(\psi_t) - F(\psi^\star) \right] - \frac{2\eta_t}{M} \sum_{m \in [M]} \mathbb{E} \langle g_t^m, \psi_t^m - \psi^\star \rangle$$

$$+ \frac{2\eta_t}{M} \sum_{m \in [M]} \langle g_t^m, \psi_t^m - \psi_t \rangle + \frac{2\eta_t^2 \gamma \sigma^2}{M},$$

$$\leq \mathbb{E} \|\psi_t - \psi^\star\|_H^2 + 2\eta_t^2 L^2 (1 + \alpha) \mathbb{E} \xi_t - \frac{2\eta_t}{M} \sum_{m \in [M]} \mathbb{E} \left[ \hat{F}_m(w_t^m, \theta_t^m) - \hat{F}_m(w^\star, \theta_m^\star) \right]$$

$$+ \frac{2\eta_t}{M} \sum_{m \in [M]} \mathbb{E} \left[ \hat{F}_m(w_t^m, \theta_t^m) - \hat{F}_m(w_t, \theta_t^m) + \frac{L}{2} \|w_t^m - w_t\|^2 \right]$$

$$+ 2\eta_t^2 L(1 + \alpha) \mathbb{E} \left[ \hat{F}(\psi_t) - \hat{F}(\psi^\star) \right] + \frac{2\eta_t^2 \gamma \sigma^2}{M},$$

$$= \mathbb{E} \|\psi_t - \psi^\star\|_H^2 + \left( 2\eta_t^2 L^2 (1 + \alpha) + \eta_t L \right) \mathbb{E} \xi_t - \left( 2\eta_t - 2\eta_t^2 L(1 + \alpha) \right) \cdot \mathbb{E} \left[ \hat{F}(\psi_t) - \hat{F}(\psi^\star) \right]$$

$$+ \frac{2\eta_t^2 \gamma \sigma^2}{M},$$

$$= \mathbb{E} \|\psi_t - \psi^\star\|_H^2 + \eta_t L (1 + 2\eta_t L(1 + \alpha)) \mathbb{E} \xi_t - 2\eta_t (1 - \eta_t L(1 + \alpha)) \cdot \mathbb{E} \left[ F(\psi_t) - F(\psi^\star) \right]$$

$$+ \frac{2\eta_t^2 \gamma \sigma^2}{M},$$

$$\leq \mathbb{E} \|\psi_t - \psi^\star\|_H^2 + 2\eta L \mathbb{E} \xi_t - \eta_t \cdot \mathbb{E} \left[ F(\psi_t) - F(\psi^\star) \right] + \frac{2\eta_t^2 \gamma \sigma^2}{M},$$

where we assume $\eta_t \leq \frac{1}{2L(1+\alpha)}$. Re-arranging this, we get,

$$\mathbb{E} \left[ F(\phi_t) - F(\phi^\star) \right] \leq \frac{1}{\eta_t} \left( \mathbb{E} \|\phi_t - \phi^\star\|_H^2 - \mathbb{E} \|\phi_{t+1} - \phi^\star\|_H^2 \right) + 2L \mathbb{E} \xi_t + \frac{2\eta_t \gamma \sigma^2}{M}.$$

Choose $\eta_t = \eta$, then average this over time. Using the convexity of $F$, and using that, we initialize at zero, we get that for $\alpha > 0$,

$$\mathbb{E} \left[ F \left( \frac{1}{T} \sum_{t=0}^{T-1} \phi_t \right) - F^\star \right] \leq \frac{\|\phi^\star\|_H^2}{\eta T} + \frac{2L}{T} \sum_{t=0}^{T-1} \mathbb{E} \xi_t + \frac{2\eta \gamma \sigma^2}{M},$$

$$= \frac{\|w^\star\|^2 + \frac{1}{\alpha M} \sum_{m \in [M]} \|\theta_m^\star\|^2}{\eta T} + \frac{2L}{T} \sum_{t=0}^{T-1} \mathbb{E} \xi_t + \frac{2\eta \gamma \sigma^2}{M},$$

$$= \frac{\frac{1}{M} \sum_{m \in [M]} \left( \|w^\star\|^2 + \frac{1}{\alpha} \|\theta_m^\star\|^2 \right)}{\eta T} + \frac{2L}{T} \sum_{t=0}^{T-1} \mathbb{E} \xi_t + \frac{2\eta \gamma \sigma^2}{M}, \quad (104)$$

First, we note that when $\alpha = 0$, we don't lift the dimension, and then we recover the usual local SGD upper bound (Woodworth et al., 2020b). Now, we recall that the above upper bound holds for any parameterization of any $\hat{F}$ optima. Thus, to minimize the numerator in the first term, we want to minimize the quantity $\frac{1}{M} \sum_{m \in [M]} \left( \|w^\star\|^2 + \frac{1}{\alpha} \|v_m^\star - w^\star\|^2 \right)$ for any given $v_m^\star \in \arg\min_{v \in \mathcal{W}} F_m(v)$. A simple calculation shows that this is minimized when $w^\star = \frac{\frac{1}{M} \sum_{m \in [M]} v_m^\star}{1 + \alpha}$. Using this choice, we can bound the numerator in the first term of (104) as follows,

$$\frac{1}{M} \sum_{m \in [M]} \left( \|w^\star\|^2 + \frac{1}{\alpha} \|\theta_m^\star\|^2 \right) = \frac{1}{M} \sum_{m \in [M]} \left( \|w^\star\|^2 + \frac{1}{\alpha} \|v_m^\star - w^\star\|^2 \right),$$

$$\leq \frac{1}{M} \sum_{m \in [M]} \left( \left( 1 + \frac{2}{\alpha} \right) \|w^\star\|^2 + \frac{2}{\alpha} \|v_m^\star\|^2 \right),$$

$$\leq \frac{1}{M} \sum_{m \in [M]} \left( \left(1 + \frac{2}{\alpha}\right) \frac{B^2}{(1+\alpha)^2} + \frac{2}{\alpha} B^2 \right),$$

$$\leq \left( \left(1 + \frac{2}{\alpha}\right) \frac{1}{(1+\alpha)^2} + \frac{2}{\alpha} \right) B^2,$$

$$\leq 5B^2 \frac{1+\alpha}{\alpha},$$

where we used Assumption 2. Replacing this in bound (104) we get the following final result for $\eta \leq \frac{1}{2L(1+\alpha)}$,

$$\mathbb{E}\left[ F\left( \frac{1}{T} \sum_{t=0}^{T-1} \phi_t \right) - F^\star \right] \leq \frac{5B^2(1+\alpha)}{\eta \alpha T} + \frac{2\eta \gamma \sigma^2}{M} + \frac{2L}{T} \sum_{t=0}^{T-1} \mathbb{E}\xi_t,$$

which finishes the proof. $\qquad \square$

### B.5  ALTERNATE PROOF OF THEOREM 2

*Proof.* Throughout the proof, we will assume $\mathcal{W} = \mathbb{R}^d$. We will first define the following lifted vectors that will be useful for our analysis:

$$w_t := \frac{1}{M} \sum_{m \in [M]} w_t^m, \ \phi_t^m = \begin{bmatrix} w_t^m \\ \theta_t^1 \\ \vdots \\ \theta_t^M \end{bmatrix}, \ \phi_t = \frac{1}{M} \sum_{m \in [M]} \phi_t^m = \begin{bmatrix} w_t \\ \theta_t^1 \\ \vdots \\ \theta_t^M \end{bmatrix}.$$

Next, we define,

$$H = \begin{bmatrix} I_d & 0 \\ 0 & \frac{1}{M\alpha} I_{Md} \end{bmatrix}, \text{ and } H^{-1} = \begin{bmatrix} I_d & 0 \\ 0 & M\alpha I_{Md} \end{bmatrix}.$$

Note that if $\alpha = 0$, we will reduce to the non-personalized setting, and we can let $H = H^{-1} = I_d$ and not lift the dimension of the parameter space, i.e., $\phi = w \in \mathcal{W}$. We also define the following gradient for all $m \in [M]$ by lifting the dimension of gradient of $F_m$,

$$g_t^m = \nabla_\phi \hat{F}_m(w_t^m, \theta_t^m) = \begin{bmatrix} \nabla_w f_m(w_t^m + \theta_t^m; z_t^m) \\ 0 \\ \vdots \\ \nabla_\theta f(w_t^m + \theta_t^m; z_t^m) \\ \vdots \\ 0 \end{bmatrix}$$

The following gradient will also be useful,

$$g_t = \frac{1}{M} \sum_{m \in [M]} g_t^m = \begin{bmatrix} \frac{1}{M} \sum_{m \in [M]} \nabla_w f(w_t^m + \theta_t^m; z_t^m) \\ \frac{1}{M} \nabla_\theta f(w_t^1 + \theta_t^1; z_t^1) \\ \vdots \\ \frac{1}{M} \nabla_\theta f(w_t^M + \theta_t^M; z_t^M) \end{bmatrix}.$$

As well as its conditional expectation w.r.t. the filtration $\mathcal{H}_t = \sigma\left(w_t^1, \ldots, w_t^M, \theta_t^1, \ldots, \theta_t^M\right)$,

$$h_t = \mathbb{E}[g_t | \mathcal{H}_t] = \begin{bmatrix} \frac{1}{M} \sum_{m \in [M]} \nabla_w F_m(w_t^m + \theta_t^m) \\ \frac{1}{M} \nabla_\theta F_1(w_t^1 + \theta_t^1) \\ \vdots \\ \frac{1}{M} \nabla_\theta F_M(w_t^M + \theta_t^M) \end{bmatrix}.$$

We can note the following about this vector,

$$\mathbb{E}\left[ \|g_t - h_t\|_{H^{-1}}^2 \, | \, \mathcal{H}_t \right] = \mathbb{E}\left[ \left\| \frac{1}{M} \sum_{m \in [M]} \nabla_w f(w_t^m + \theta_t^m; z_t^m) - \nabla_w F_m(w_t^m + \theta_t^m) \right\|^2 \right.$$

$$+ \alpha M \sum_{m \in [M]} \frac{1}{M^2} \left\| \nabla_\theta f(w_t^m + \theta_t^m; z_t^m) - \nabla_\theta F_m(w_t^m + \theta_t^m) \right\|^2 |\mathcal{H}_t \Bigg],$$

$$= \frac{1}{M^2} \sum_{m \in [M]} \mathbb{E} \left[ \left\| \nabla_w f(w_t^m + \theta_t^m; z_t^m) - \nabla_w F_m(w_t^m + \theta_t^m) \right\|^2 \right]$$

$$+ \frac{\alpha}{M} \sum_{m \in [M]} \mathbb{E} \left[ \left\| \nabla_\theta f(w_t^m + \theta_t^m; z_t^m) - \nabla_\theta F_m(w_t^m + \theta_t^m) \right\|^2 |\mathcal{H}_t \right],$$

$$\leq \frac{\sigma^2}{M} + \alpha \sigma^2.$$

Defining $\gamma = \max\{1, \alpha M\}$ we can re-write this as,

$$\mathbb{E} \left[ \|g_t - h_t\|_{H^{-1}}^2 |\mathcal{H}_t \right] \leq 2\gamma \frac{\sigma^2}{M}. \tag{105}$$

We will use this bound later. We will also consider an arbitrary parametrization of the optima of different machines as,

$$\phi^\star = \begin{bmatrix} w^\star \\ \theta_1^\star \\ \vdots \\ \theta_M^\star \end{bmatrix} \in \mathcal{W}^{M+1}, \ w^\star + \theta_1^\star \in \arg\min_{v \in \mathcal{W}} F_m(v), \ F^\star := \hat{F}(\phi^\star).$$

Since this parameterization is arbitrary, we can choose the minimum-norm parameterization (in any norm). We will use this fact later. With this disclaimer, we can write for $\alpha > 0$,

$$\phi_{t+1} = \phi_t - \eta_t H^{-1} g_t.$$

Subtracting $\phi^\star$ and taking the norm induced by $H$, we can write the following recurrence by defining $\gamma = \max\{1, \alpha M\}$,

$$\mathbb{E} \left[ \|\phi_{t+1} - \phi^\star\|_H^2 |\mathcal{H}_t \right] = \mathbb{E} \left[ \left\| \phi_t - \eta_t H^{-1} g_t - \phi^\star \right\|_H^2 |\mathcal{H}_t \right],$$

$$= \|\phi_t - \phi^\star\|_H^2 + \eta_t^2 \mathbb{E} \left[ \|g_t\|_{H^{-1}}^2 |\mathcal{H}_t \right] - 2\eta_t \langle \mathbb{E}(g_t|\mathcal{H}_t), \phi_t - \phi^\star \rangle,$$

$$= \|\phi_t - \phi^\star\|_H^2 + \eta_t^2 \mathbb{E} \left[ \|g_t - h_t + h_t\|_{H^{-1}}^2 |\mathcal{H}_t \right] - 2\eta_t \langle h_t, \phi_t - \phi^\star \rangle,$$

$$= \|\phi_t - \phi^\star\|_H^2 + \eta_t^2 \mathbb{E} \left[ \|g_t - h_t\|_{H^{-1}}^2 |\mathcal{H}_t \right] + \eta_t^2 \mathbb{E} \left[ \|h_t\|_{H^{-1}}^2 |\mathcal{H}_t \right] - 2\eta_t \langle h_t, \phi_t - \phi^\star \rangle,$$

where we used the fact that the cross terms are zero because $h_t$ is measurable under $\mathcal{H}_t$ and $h_t = \mathbb{E}[g_t|\mathcal{H}_t]$. Defining $h_t^m = \mathbb{E}[g_t^m|\mathcal{H}_t]$, so that $h_t = \frac{1}{M} \sum_{m \in [M]} h_t^m$ and using the variance bound in (105) we get that,

$$\mathbb{E} \left[ \|\phi_{t+1} - \phi^\star\|_H^2 |\mathcal{H}_t \right] \leq \|\phi_t - \phi^\star\|_H^2 + 2\gamma \eta_t^2 \frac{\sigma^2}{M} + \eta_t^2 \left\| \frac{1}{M} \sum_{m \in [M]} \nabla_w F_m(w_t^m + \theta_t^m) \right\|^2$$

$$+ \eta_t^2 \alpha M \sum_{m \in [M]} \frac{1}{M^2} \|\nabla F_m(w_t^m + \theta_t^m)\|^2 - \frac{2\eta_t}{M} \sum_{m \in [M]} \langle h_t^m, \phi_t - \phi^\star \rangle,$$

$$\leq \|\phi_t - \phi^\star\|_H^2 + 2\gamma \eta_t^2 \frac{\sigma^2}{M} + \frac{\eta_t^2(1 + \alpha)}{M} \sum_{m \in [M]} \|\nabla_w F_m(w_t^m + \theta_t^m)\|^2$$

$$- \frac{2\eta_t}{M} \sum_{m \in [M]} \left( \langle \nabla F_m(w_t^m + \theta_t^m), w_t - w^\star \rangle + \langle \nabla F_m(w_t^m + \theta_t^m), \theta_t^m - \theta_m^\star \rangle \right),$$

$$\leq \|\phi_t - \phi^\star\|_H^2 + 2\gamma \eta_t^2 \frac{\sigma^2}{M} + \frac{2L\eta_t^2(1 + \alpha)}{M} \sum_{m \in [M]} (F_m(w_t^m + \theta_t^m) - F_m^\star)$$

$$- \frac{2\eta_t}{M} \sum_{m \in [M]} \langle \nabla F_m(w_t^m + \theta_t^m), w_t + \theta_t^m - v_m^\star \rangle,$$

$$\leq \|\phi_t - \phi^\star\|_H^2 + 2\gamma\eta_t^2 \frac{\sigma^2}{M} + \frac{2L\eta_t^2(1+\alpha)}{M} \sum_{m \in [M]} (F_m(w_t^m + \theta_t^m) - F_m^\star)$$

$$- \frac{2\eta_t}{M} \sum_{m \in [M]} (F_m(w_t^m + \theta_t^m) - F_m^\star),$$

$$= \|\phi_t - \phi^\star\|_H^2 + 2\gamma\eta_t^2 \frac{\sigma^2}{M} - \left(2\eta_t - 2(1+\alpha)L\eta_t^2\right) \frac{1}{M} \sum_{m \in [M]} (F_m(w_t^m + \theta_t^m) - F_m^\star),$$

$$\leq \|\phi_t - \phi^\star\|_H^2 + 2\gamma\eta_t^2 \frac{\sigma^2}{M} - \frac{\eta_t}{M} \sum_{m \in [M]} (F_m(w_t^m + \theta_t^m) - F_m^\star),$$

where we used that $\eta_t \leq \frac{1}{2L(1+\alpha)}$. Re-arranging this using $\eta_t = \eta$, we get that,

$$\frac{1}{M} \sum_{m \in [M]} (F_m(w_t^m + \theta_t^m) - F_m^\star) \leq \frac{\|\phi_t - \phi^\star\|_H^2 - \|\phi_{t+1} - \phi^\star\|_H^2}{\eta} + 2\gamma\eta\frac{\sigma^2}{M}.$$

Averaging both sides over time and using the convexity of $F_m$'s we get that.

$$\frac{1}{M} \sum_{m \in [M]} \left(F_m\left(\frac{1}{T}\sum_{t=0}^{T-1} w_t^m + \theta_t^m\right) - F_m^\star\right) \leq \frac{\|\phi^\star\|_H^2}{\eta T} + 2\gamma\eta\frac{\sigma^2}{M}. \tag{106}$$

Now, we recall that the above upper bound holds for any parameterization of any $\hat{F}$ optima. Thus, to minimize the numerator in the first term, we want to minimize the quantity $\frac{1}{M} \sum_{m \in [M]} \left(\|w^\star\|^2 + \frac{1}{\alpha} \|v_m^\star - w^\star\|^2\right)$ for any given $v_m^\star \in \arg\min_{v \in \mathcal{W}} F_m(v)$. A simple calculation shows that this is minimized when $w^\star = \frac{\frac{1}{M} \sum_{m \in [M]} v_m^\star}{1+\alpha}$. Using this choice, we can bound the numerator in the first term of (104) as follows,

$$\frac{1}{M} \sum_{m \in [M]} \left(\|w^\star\|^2 + \frac{1}{\alpha} \|\theta_m^\star\|^2\right) = \frac{1}{M} \sum_{m \in [M]} \left(\|w^\star\|^2 + \frac{1}{\alpha} \|v_m^\star - w^\star\|^2\right),$$

$$\leq \frac{1}{M} \sum_{m \in [M]} \left(\left(1 + \frac{2}{\alpha}\right) \|w^\star\|^2 + \frac{2}{\alpha} \|v_m^\star\|^2\right),$$

$$\leq \frac{1}{M} \sum_{m \in [M]} \left(\left(1 + \frac{2}{\alpha}\right) \frac{B^2}{(1+\alpha)^2} + \frac{2}{\alpha}B^2\right),$$

$$\leq \left(\left(1 + \frac{2}{\alpha}\right) \frac{1}{(1+\alpha)^2} + \frac{2}{\alpha}\right) B^2,$$

$$\leq 5B^2 \frac{1+\alpha}{\alpha},$$

where we used Assumption 2. Replacing this in bound (106), we get the following final result for $\eta \leq \frac{1}{2(1+\alpha)L}$,

$$\frac{1}{M} \sum_{m \in [M]} \left(F_m\left(\frac{1}{T}\sum_{t=0}^{T-1} w_t^m + \theta_t^m\right) - F_m^\star\right) \leq \frac{5B^2(1+\alpha)}{\eta\alpha T} + 2\eta\frac{\sigma^2}{M}\max\{1, \alpha M\},$$

which finishes the proof. □

### B.6 PROOF OF PROPOSITION 1

*Proof.* To elucidate this, we consider the following simple problem with $L = 1$ in a single dimension with two clients, i.e., $\mathcal{W} = \mathbb{R}$:

$$F_1(v) = \frac{1}{2}\left(v - (v^\star - \zeta^\star)^2\right),$$

$$F_2(v) = \frac{1}{2} \left( v - (v^\star + \zeta^\star) \right)^2, \tag{107}$$

$$F(v) = \frac{F_1(v) + F_2(v)}{2},$$

$$= \frac{1}{2} (v - v^\star)^2 + \frac{\zeta_\star^2}{2L}, \tag{108}$$

We note that this problem satisfies Assumption 1 with $\mu = 0$ as the client objectives have directions of no information. We will also assume that the $\zeta_\star$ and $v^\star \in \mathbb{R}^3$ are such that the problem satisfies Assumption 2. Finally, we note that the objective satisfied the bounded heterogeneity Assumption 3. Using more or less the same series of calculations as in the proof of Theorem 1 in the noiseless setting, one can derive the sequence of iterates of the global and the personal models. In particular, following similar steps, one gets the following form for the consensus error,

$$\frac{1}{2} \sum_{m \in [2]} \left\| w_{t+1}^m - w_{t+1} \right\|^2 = \left( \frac{\alpha \nu^K + 1}{\alpha + 1} \right)^{\tau(t)/K} \cdot \frac{\zeta_\star^2}{1 + \alpha} \cdot \left( 1 - \nu^{t-1-\tau(t)} \right),$$

where $\tau(t)$ is the last time smaller than or equal to $t$ when communication happened and $\nu := 1 - \eta(1 + \alpha)$. Simplifying this and averaging it over time gives the desired result in the proposition. $\square$

