# OpenReview forum: "Personalization Mitigates the Perils of Local SGD for Heterogeneous Distributed Learning"
_ICLR.cc/2024/Conference — ICLR 2024 Conference Withdrawn Submission_

### Official Review · Reviewer_PXND · 2023-10-29

**Soundness:** 2 fair
**Presentation:** 2 fair
**Contribution:** 2 fair
**Rating:** 3
**Confidence:** 3

**Summary:**

The paper studied a family of personalized versions of local stochastic gradient descent. They establish convergence rates for the strongly convex problem and identify certain hyperparameter regimes where the proposed algorithm outperforms the purely local training.

**Strengths:**

The paper is well-written and easy to follow.

The problem studied is interesting and related to federated learning.

The key message that over-parameterization is helpful to convergence for personalized models is indeed interesting and new to me.

**Weaknesses:**

1. The support for the benefit of over-parameterization is not sufficient. The paper lacks experiments which makes the theoretical claims less convincing.

2. There are some overclaims and unfair comparisons.

3. There is no empirical investigation for the convergence of convex cases.

**Questions:**

1.  How to interpret (2) where we want to minimize a function taking the value of a $M$-dimensional vector? Note that we can’t compare any two multidimensional vectors. I think the formulation in (2) is not rigorous and strange.

2. I think (5) is the same as (1). If $v_m^{\star}$ is the minimizer of the $m$-th objective function $F_m$, then (v_1^{\star}, \cdots, v_M^{\star}) is the exact solution of (1). Then $\theta_m = v_m^{\star} - \bar{v}^{\star}$ and $w = \bar{v}^{\star} := \frac{1}{K} \sum_{m=1}^K v_m^{\star}$ is the minimizer of (5). Hence, the introduction of $w$ is useless at first glance. In other words, the introduction of $w$ didn’t change the problem we wanted to solve but introduced additional complexity to optimize it. As a result, there will be infinitely many pairs of solutions minimizing the problem (5). This property is named over-parameterization in the paper.

3. The author claimed that such an ``over-parameterization’’ is beneficial in enlarging the solution regime. However, I’m not convinced that the additional $w$ is beneficial. The paper considers a convex optimization problem, so the optimal loss value is unique (the optimal solution might not). Also, keep in mind that the paper still wants to find all local minimizers rather than a global minimizer. So I believe the pure local training is the optimal algorithm. Note that in Theorem 1, the author identified the regimes where the proposed algorithm outperforms pure local training. The main advance is that the noise terms are scaled by the number of devices for the proposed method (from Table 1). To guarantee this advantage, a restriction on the local update $K$ is posed. For me, it is more like a mathematical trick rather than an algorithmic advantage.  Furthermore, no experiments/simulation results are provided, it is harder for me to believe the introduction of $w$ to the original problem (1) and the proposed method is indeed better than pure local training and this outperformance is due to algorithmic development rather than mathematical tricks.

4. Given that (5) is equal to (1), some comparisons in the paper are actually not very fair.  For example, at the first point of contributions (on Page 3), the author said ``We make no data heterogeneity assumptions in our analysis, which is a significant improvement over every known vanilla local SGD analysis.’’. The comparison is unfair, because most local methods try to find a consensus model, while here in this paper, the goal is to find personalized models. Hence, to make each personalized model converge, you of course do not need any heterogeneity assumptions.

---

### Official Review · Reviewer_bZrV · 2023-10-30

**Soundness:** 3 good
**Presentation:** 4 excellent
**Contribution:** 3 good
**Rating:** 8
**Confidence:** 2

**Summary:**

In the work "Personalization Mitigates the Perils of Local SGD for Heterogeneous Distributed learning" the authors propose a model problem for personalized learning problem and then a scheme to find the solution to the introduced problem. The introduced problem allows for coupling amongst each agents' task while at the same time introducing a model parameter that is only relevant to each individual agent. This is done via a very simple parametrization that yields as an advantage that computing gradients is cheap.
The authors then provide convergence guarantees and demonstrate the scheme's advantages when compared to SGD.

**Strengths:**

The paper proposes a very convenient model for personalized learning that has algorithmic advantages. Additionally the authors can provide algorithmic guarantees for the scheme, at least for the strongly convex and smooth case, that match what would be expected.

The paper is overall very well written and in terms of significance, while I am not familiar with the personalized learning literature, given the simplicity of the model I believe it can inspire other works.

**Weaknesses:**

See questions.

**Questions:**

- The introduction of personalized learning is partially justified by the need of correcting for gradient heterogeneity in a federated learning context. Could this not be alleviated instead via gradient correction, i.e. via some gradient tracking scheme?
- This leads to my next question. In case the above issue can be alleviated via some gradient correction mechanism, while the proposed model is still interesting in case we are trying to obtain a more personalized solution, is there any application for which this problem formulation (or a very closely related one) comes naturally?

---

### Official Review · Reviewer_r2Eb · 2023-10-31

**Soundness:** 3 good
**Presentation:** 2 fair
**Contribution:** 1 poor
**Rating:** 3
**Confidence:** 3

**Summary:**

This paper investigates a personalized version of FedAvg in which there is a shared parameter among clients, and each client possesses a personalized parameter. The final model for each client is determined by the summation of these parameters. The authors analyze the convergence of this model for non-i.i.d. clients without any additional assumptions regarding data heterogeneity for strongly convex functions.

**Strengths:**

1) Dealing with data heterogeneity between non-i.i.d. clients is an important problem for the community. Building a connection between the commonly used assumption on data heterogeneity and personalized FL is a very interesting topic.

2) The authors prove the convergence of their method for strongly convex functions without any assumption.

**Weaknesses:**

1) In practice for FL problems the *non-convex* neural networks are used, however the paper studies strongly convex optimization, and for general convex functions a not-proved conjecture is provided.

2) The additive model of shared and personalized parameter without any algorithmic limitations is not a reasonable algorithm for personalized FL problems, especially, in the strongly convex setting, which the final models will be the global minimum of each clients.
Usually, in FL the clients don't have many samples, and an important point in personalized FL, is to use the shared information between different clients to find models with better generalizations.
The studied model here, doesn't have this property, and I don't see that the results would still hold for any extensions of it, since they uses convergence to the global minimum of each clients.

3) Lack of any experiments to show the performance of the algorithm in practice to validate the results.

4) Studying the trade-offs between convergence and generalization throughout the training would be an interesting experiment.

**Questions:**

Regarding the weakness 2 and issues with generalization, can you extend the results to scenarios which the shared weight contains some shared information and the personalized parameters complete it? (For example the case that the norm of personalized parameters is bounded.)

---

### Official Review · Reviewer_JN1e · 2023-11-01

**Soundness:** 1 poor
**Presentation:** 2 fair
**Contribution:** 2 fair
**Rating:** 1
**Confidence:** 4

**Summary:**

The paper considers the problem of finding a fully personalized model for each client participating in the distributed training process. In particular, the authors propose a reformulation of the fully personalized problem and adjust Local-SGD/FedAvg method for such a problem. The main idea is to introduce an extra variable for each device, use different stepsizes for each variable during the local steps, and aggregate the global variable only. The authors analyze the proposed method for smooth strongly convex and convex problems.

**Strengths:**

S1. The authors clearly explain the idea behind the method.

**Weaknesses:**

W1. The paper makes several inaccurate claims that may mislead a non-expert reader. I provide them in the section with questions and comments.

W2. Theorem 1 has several issues.

- W2.1. The step in the proof from formula (52) to formula (53) is incorrect: the authors claim that it follows from the independence of the samples used on each machine at each time step, but this does not imply independence of $\delta^r$ and $G^r$. Indeed, if the workers do strictly more than 1 local step, then $G^r$ depends on the random iterates generated through the round $r$ that depends on the stochasticity from $\delta^r$.

- W2.2. The result of the theorem is not better than for the pure local training. Indeed, for the pure local training one can get from the standard analysis of SGD [1] the following upper bound: $\exp(-\eta \mu T) B^2 + \frac{\eta \sigma^2}{\mu}$, where $T$ is the total number of steps (corresponds to $T=  KR$ fro Local-SGD). The upper bound from formula (15) is $\exp(-R(1 - e^{-2\eta \mu K})) B^2 + \frac{8\eta \sigma^2}{\mu}$. Next, the maximum of the function $g(x) = (1 - e^{-2x}) / x$ for $x \in [0,1]$ equals $2$ (attained at $x = 0$ if we continuously extend the function at zero). This means that for small enough values of $\eta$, which are considered in the theorem and described in Section B.2, we have that $R(1-e^{2\eta \mu K}) \leq 2\eta\mu KR = 2\eta\mu T$. Therefore, for the SGD without synchronization (pure local training) we have the following upper bound: $\exp(-R(1 - e^{-2\eta \mu K})/2) B^2 + \frac{\eta \sigma^2}{\mu}$. This upper bound has two times worse exponent in the first term and 8 times better second term than in formula (15). Therefore, in the same settings as for formula (16), purely local SGD has upper bound $\exp(-KR/(2\kappa)) B^2 + \frac{\sigma^2}{8\mu^2 MKR}$, which is even better than (16) when the second term is dominating, which is typically the case. That is, **Theorem 1 shows no benefit of the considered method over the fully local training**.

W3. Theorem 2 also has some issues.

- W3.1. The upper bound for the third term in (17) depends on the heterogeneity of the problem. If one does only local SGD steps and never runs communication rounds, then one can get a similar result to (18).

- W3.2. The result from (18) is not better than for the standard SGD run on each machine locally (without communications). That is, **Theorem 2 shows no benefit of the considered method over the fully local training**.

W4. The paper provides no numerical experiments with the comparison to other methods for PFL. In view of the above issues, this significantly weakens the contribution of the paper.

[1] Gower, R. M., Loizou, N., Qian, X., Sailanbayev, A., Shulgin, E., & Richtárik, P. (2019, May). SGD: General analysis and improved rates. In International conference on machine learning (pp. 5200-5209). PMLR.

**Questions:**

### Main questions and comments

1. On page 3 (the top) when describing the approaches for problem (4) the authors claim that $g : \mathcal{W}^2 \to \mathcal{W}$, which is not accurate. For example, Bietti et al. (2022) and Mishchenko et al. (2023) allow $w$ and $\theta_m$ to be different parameters of the model, e.g., the common "head" and different "tails" of the neural network.

2. In view of the above, in problem (4) the solutions can be different form the ones in (11). Therefore, the remark after (11) should be fixed.

3. The discussion after (12) is a bit messy: what is meant there by function sub-optimality?

4. Page 7, last sentence before Section 3.1: Mishchenko et al. (2023) also provide theoretical results showing the positive impact of personalization.

5. Page 7, "we make no heterogeneity assumptions meaning this upper bound is better for high enough heterogeneity than all federated learning algorithms": this is not true due to W2 and W3 + there are algorithms such as SCAFFOLD (Karimireddy et al., 2020) that also do not depend on the heterogeneity.


### Minor comments

1. In Table 1, one needs to use $\leq$ instead of $\preccurlyeq$.

2. In (35), there should be no $\nabla$.

3. In (41), $\delta^r$ is undefined.

4. There is some typo in the formula after (42) because the mentioned product is not equal to $I$.